# Achieving functional neuronal dendrite structure through sequential stochastic growth and retraction

André Ferreira Castro[1,2,3#‡§]*, Lothar Baltruschat[3], Tomke Stürner[3,4], Amirhoushang Bahrami[5], Peter Jedlicka[1,6,7], Gaia Tavosanis[3,8†]*, Hermann Cuntz[1,2†]*

[1]Frankfurt Institute for Advanced Studies, Frankfurt am Main, Germany; [2]Ernst Strüngmann Institute (ESI) for Neuroscience in cooperation with Max Planck Society, Frankfurt am Main, Germany; [3]Center for Neurodegenerative Diseases (DZNE), Bonn, Germany; [4]Department of Zoology, University of Cambridge, Cambridge, United Kingdom; [5]Max Planck Institute for Dynamics and Self Organization, Göttingen, Germany; [6]Faculty of Medicine, ICAR3R – Interdisciplinary Centre for 3Rs in Animal Research, Justus Liebig University Giessen, Giessen, Germany; [7]Neuroscience Center, Institute of Clinical Neuroanatomy, Goethe University, Frankfurt am Main, Germany; [8]LIMES Institute, University of Bonn, Bonn, Germany

*For correspondence:
acastro@mrc-lmb.cam.ac.uk (AFC);
Gaia.Tavosanis@dzne.de (GT);
cuntz@fias.uni-frankfurt.de (HC)

†These authors contributed equally to this work
#Twitter: @AFerreiraCastro

Present address: ‡Department of Physiology, Development and Neuroscience, University of Cambridge, Cambridge, United Kingdom; §MRC Laboratory of Molecular Biology, Francis Crick Avenue, Cambridge Biomedical Campus, Cambridge, United Kingdom

Competing interests: The authors declare that no competing interests exist.

**Abstract** Class I ventral posterior dendritic arborisation (c1vpda) proprioceptive sensory neurons respond to contractions in the *Drosophila* larval body wall during crawling. Their dendritic branches run along the direction of contraction, possibly a functional requirement to maximise membrane curvature during crawling contractions. Although the molecular machinery of dendritic patterning in c1vpda has been extensively studied, the process leading to the precise elaboration of their comb-like shapes remains elusive. Here, to link dendrite shape with its proprioceptive role, we performed long-term, non-invasive, in vivo time-lapse imaging of c1vpda embryonic and larval morphogenesis to reveal a sequence of differentiation stages. We combined computer models and dendritic branch dynamics tracking to propose that distinct sequential phases of stochastic growth and retraction achieve efficient dendritic trees both in terms of wire and function. Our study shows how dendrite growth balances structure–function requirements, shedding new light on general principles of self-organisation in functionally specialised dendrites.

## Introduction

A fundamental open question in neuroscience is understanding how the shape of specific neuron classes arises during cell development to perform distinct computations (*Carr et al., 2006*). In the past, technological and conceptual advances have allowed exciting discoveries on how the coupling of class type-specific dendrite geometry with various ion channels provide the substrate for signal processing and integration in dendrites (*Mainen and Sejnowski, 1996*; *van Elburg and van Ooyen, 2010*; *Gabbiani et al., 2002*; *London and Häusser, 2005*; *Branco et al., 2010*; *Stuart and Spruston, 2015*; *Beaulieu-Laroche et al., 2018*; *Poirazi and Papoutsi, 2020*). Also, dendrite structure has been successfully linked to connectivity and wiring requirements allowing the generation of highly realistic synthetic dendritic morphologies based on these principles alone (*Stepanyants et al., 2004*; *Wen and Chklovskii, 2008*; *Cuntz et al., 2010*; *Cuntz et al., 2007*; *Nanda et al., 2018*).

However, to date, these efforts have fallen short of clarifying the link between the developmental elaboration of dendrite structure and the structural constraints dictated by the computational tasks of the neuron (*Lefebvre et al., 2015*). Unravelling these patterning processes is important to achieve a mechanistic understanding of the nervous system and to gather insights into neurological and neurodevelopmental disorders alike (*Copf, 2016*; *Real et al., 2018*; *Forrest et al., 2018*). To attain an integrative view of dendrite functional assembly we decided to analyse a genetically tractable animal model, such as *Drosophila*, with existing comprehensive research in the fields of dendrite development, structure and function. Extensive investigations in the emergence of dendritic morphology (*Jan and Jan, 2010*; *Gerhard et al., 2017*; *Schneider-Mizell et al., 2016*; *Enriquez et al., 2015*; *Corty et al., 2016*; *Ganguly et al., 2016*; *Nanda et al., 2018*; *Couton et al., 2015*; *Schlegel et al., 2017*; *Hu et al., 2020*; *Sheng et al., 2018*) and on the specific impact of dendritic morphology on computation (*Dewell and Gabbiani, 2017*; *Single and Borst, 1998*; *Gabbiani et al., 2002*; *Cuntz et al., 2003*; *Allen et al., 2006*; *Kohl et al., 2013*; *Eichler et al., 2017*; *Frechter et al., 2019*; *He et al., 2019*) make insects notably favourable to study mechanisms of development of dendrite form and function.

A set of four distinct classes of dendritic arborisation sensory neurons of the *Drosophila* larva peripheral nervous system are of particular interest because of the marked differences in their morphology and function (*Grueber et al., 2002*). Among these cell types, the function of class I (c1da) proprioceptors is thought to tightly depend on dendritic morphology. In fact, c1da dendrites undergo sequential deformation in consecutive hemisegments by the contraction of the larva body wall during crawling (*Heckscher et al., 2012*). The structural deformation of c1da terminal branches coincides with c1da $Ca^{2+}$ responses, an activation that could provide a possible proprioceptive feedback to coordinate the peristaltic waves of muscle contractions (*Hughes et al., 2007*; *Song et al., 2007*; *Vaadia et al., 2019*).

Membrane curvature during branch deformation is thought to be directly linked to the opening of mechanically gated ion channels present in the c1da neuron membrane (*He et al., 2019*). These findings are supported by previous studies, where genetic manipulation of c1da neuron morphology (*Hughes et al., 2007*; *Song et al., 2007*) or null mutations of mechanosensitive channels expressed in the membrane of these sensory neurons (*Cheng et al., 2010*; *Guo et al., 2016*) impaired the crawling behaviour. Taken together, these data suggest that the relay of proprioceptive information about body movement is crucially dependent on the specific localisation of c1da neurons in the body segments, the association of their dendrites with the larval body wall and their precise dendritic morphology (*Fushiki et al., 2016*; *Grueber et al., 2007*; *Vaadia et al., 2019*).

In particular, the dendrites of the ventral posterior c1da neuron (c1vpda) exhibit an unmistakable stereotypical comb-like shape with a main branch (MB) running perpendicularly to the anteroposterior direction of contraction and lateral branches typically running parallel to the direction of contraction. As the peristaltic muscle contraction wave progresses along the anteroposterior axis during crawling lateral branches bend, while the MB remains almost unaffected. The different deformation profiles likely arise from the distinct orientation of the branches (*Vaadia et al., 2019*).

Dendrite morphology, dendrite activation pattern and function of c1vpda neurons are known. These sensory neurons thus provide an ideal platform to address how dendrite structure is optimised towards the neuron's appropriate functional response and such an optimised structure is achieved. Do dendrites form through an intrinsic deterministic program or are they shaped by stochastic processes? Moreover, do these functional requirements coexist with optimal wire constraints, that is, minimisation of dendrite cable material costs, observed in many neuronal dendrites (*Cuntz et al., 2007*; *Wen and Chklovskii, 2008*)? In this work, we used the c1vpda neuron to address precisely these key questions. We reasoned that by elucidating the spatiotemporal differentiation of the cell we could further our understanding of how functionally constrained morphologies emerge during development. In previous studies, analysis of the underlying developmental trajectories of distinct cell types provided important insights into how neurons (*Miller, 1981*; *Lim et al., 2018*) and circuits (*Langen et al., 2015*) pattern into functional structures.

We therefore combined long-term time-lapse imaging of dendrite development, quantitative analysis, theoretical modelling, calcium imaging in freely moving animals, and *in silico* morphological modelling to describe the spatiotemporal patterning of c1vpda dendrites. We find that dendrite growth can to a large degree be described by a random growth process that satisfies optimal wire and a randomised retraction of branches that preferentially preserves functional dendrites.

## Results

### Embryonic and larval differentiation of c1vpda dendrites

To better understand the relationship between dendrite structure and function in c1vpda sensory neurons, we dissected the developmental process of apical dendrite formation quantitatively using long-term, non-invasive time-lapse imaging from embryonic stages (16 hrs after egg laying AEL) until early 3rd larval stage (72 hrs AEL) (*Figure 1*).

To visualise cell morphology we expressed a membrane-tagged fluorescent protein specifically in c1vpda neurons. Within the egg (*Figure 1A*), the main branch (MB) emerged from the soma at around 16 hrs AEL and extended in a dorsal orientation. Afterwards, a number of second-order lateral interstitial branches appeared from the initial MB extending in both the anterior and posterior directions, with the MB dorsal position potentially biasing their growth direction along the

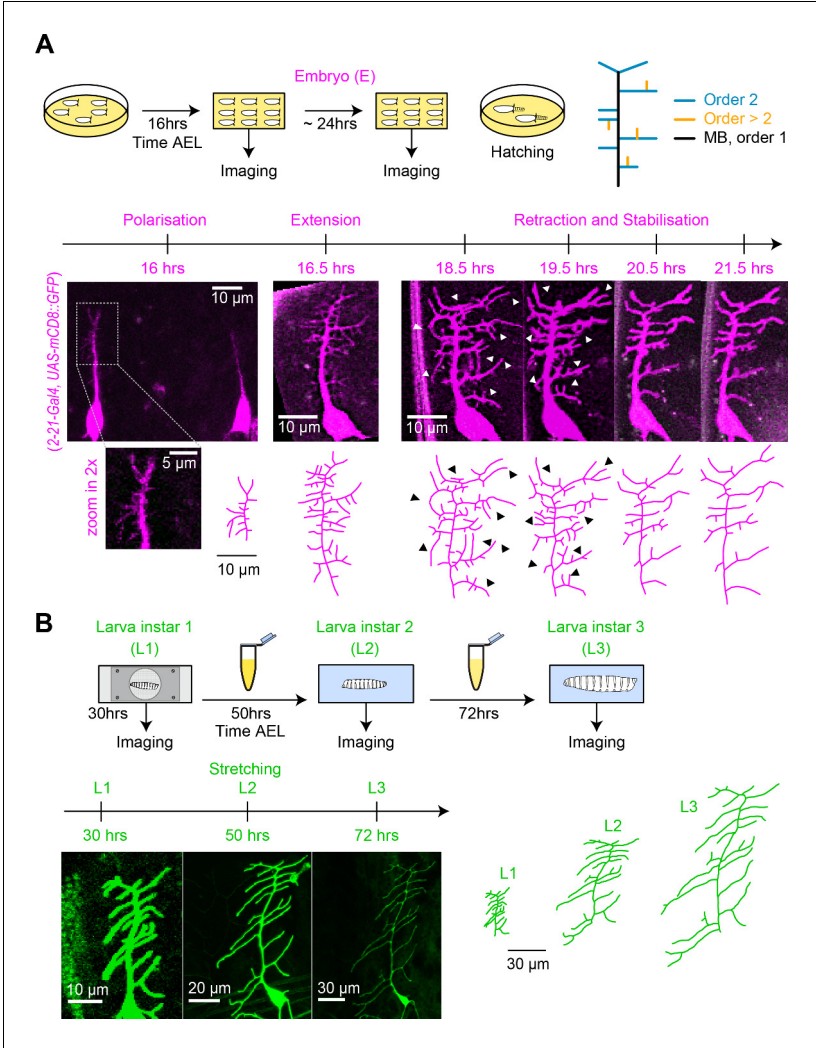

**Figure 1.** Distinct stages of c1vpda dendrite differentiation during embryonic and larval stages. (**A**) Imaging procedure throughout embryonic (E) stages. The eggs were imaged at higher temporal resolution in a time window ranging from 16−24 hrs AEL. Sketch (top row left) illustrating the experimental conditions, drawing (top row right) depicting the ordering of c1vpda branches (black: MB order 1, blue: lateral branch order 2, orange: lateral branch order > 2). Timeline and maximum intensity projections (middle row) of image stacks as well as reconstructions (bottom row) of a given representative c1vpda dendrite. White arrows in images and corresponding black arrows on reconstructions indicate exemplary changes between the time points (see main text). (**B**) Subsequent imaging of Larval instar (L) 1, 2, 3 stages with similar arrangements as in A. Times shown are AEL (after egg laying).

anteroposterior axis (*Yoong et al., 2019*). Then, shorter third-order lateral branches sprouted interstitially from the second-order lateral branches mainly along the dorsoventral axis. Lateral branches underwent repeated cycles of extension and retraction until reaching a maximum number of branches around 18.5−19 hrs AEL. Even at this stage few fourth or fifth-order lateral branches were observed.

The c1vpda sensory neuron then entered a stage of arbour reorganisation, marked predominantly by the retraction of branch tips (*Figure 1A*, 18.5 hrs and 19.5 hrs). This phase of removal of dendritic branches, hereafter referred as the retraction phase, was followed by a pre-hatching stabilisation period (*Figure 1A*). During hatching, larvae showed severe head swings and anteroposterior contractions, followed by body swirls inside the egg preventing the collection of images in this period.

After hatching (24 hrs AEL), we imaged dendrite development at the time points of 30 hrs, 50 hrs and 72 hrs AEL (*Figure 1B*). The neurons continued growing concomitantly with the expansion of the body wall. However, the post-embryonic growth phase preserved the shape and complexity of c1vpda dendrites, with only very few new branches emerging. The increase of dendrite cable was due primarily to the scaling elongation of existing branches. The dendritic pattern observed at 30 hrs AEL was fundamentally the same as the one observed at 72 hrs AEL, consistent with an isometric scaling of da sensory neurons during larval stages (*Parrish et al., 2009*).

To gain a quantitative insight into the morphological maturation process of these sensory neurons, we reconstructed the dendrites in the image stacks obtained from the time-lapse imaging and we measured their structure using 49 distinct morphometrics (see Materials and methods). Using a t-distributed Stochastic Neighbour Embedding (tSNE) (*van der Maaten, 2008*) of the entire dataset we reduced the 49—dimensional space to a 2D plot preserving neighbourhood relationships that indicate morphological similarity (*Figure 2A*). After examining the tSNE plot, it is evident that developmental time was a strong source of variation in the data with neurons becoming increasingly morphologically divergent over time. Cells from early stages formed large continuums in the tSNE plot, whereas darker green discrete clusters emerged at later stages (50−72 hrs AEL) due to dendrite morphological dissimilarity between the observation points. The non-linear developmental trajectory (yellow arrow) in the early embryonic stages reflects the intense dynamics of arbour outgrowth and refinement, while the subsequent more linear trajectory corresponds to the isometric stretching occurring in later stages.

These observations were in line with the individual developmental trajectories of number of branch points (*Figure 2B*), total dendrite length and 2D surface area of the dendritic spanning field (*Figure 2C*). To further quantify the differentiation process of c1vpda neurons, we compared the relationships of these morphometrics across the different developmental phases (*Figure 2—figure supplement 1*). During the initial extension phase, new branches were added with a linear increase with total length ($R^2 = 0.86$) and surface area alike ($R^2 = 0.73$; *Figure 2—figure supplement 1A*). Accordingly, the dendrite cable length also increased linearly with the available spanning area ($R^2 = 0.92$; *Figure 2—figure supplement 1A*).

Throughout the retraction phase, the dendrite cable length decreased linearly with the reduction of branches ($R^2 = 0.77$; *Figure 2—figure supplement 1B*). However, the retraction of branches only slightly affected the surface area of the cell ($R^2 = 0.21$), neither did the reduction of dendrite cable ($R^2 = 0.41$; *Figure 2—figure supplement 1B*). This suggests that shorter, proximally located, higher-order lateral branches (third order or higher) were the ones most strongly involved in retraction (see also *Figure 1A*, arrows). These branches, due to their location in the inner part of the dendritic field had only a small influence in defining the spanning area of the c1vpda dendrites.

In the subsequent stabilisation phase, virtually no new branches were added despite of the small increase of the total length ($R^2 = 0.33$) and surface area ($R^2 = 0.27$; *Figure 2—figure supplement 1C*). Dendrite cable length slightly increased linearly with the available spanning area ($R^2 = 0.74$; *Figure 2—figure supplement 1C*), but at a lower rate than during the initial extension phase.

Finally, only very few new branches emerged during the stretching phase from c1vpda dendrites regardless of the increase of dendrite cable ($R^2 = 0.17$), or new available surface area ($R^2 = 0.1$; *Figure 2—figure supplement 1D*). Dendrite cable length increased linearly with the available spanning area ($R^2 = 0.97$; *Figure 2—figure supplement 1D*).

Comparing the relationships between basic geometric features of tree structures has previously allowed linking dendritic architecture with wire saving algorithms (*Cuntz et al., 2012*;

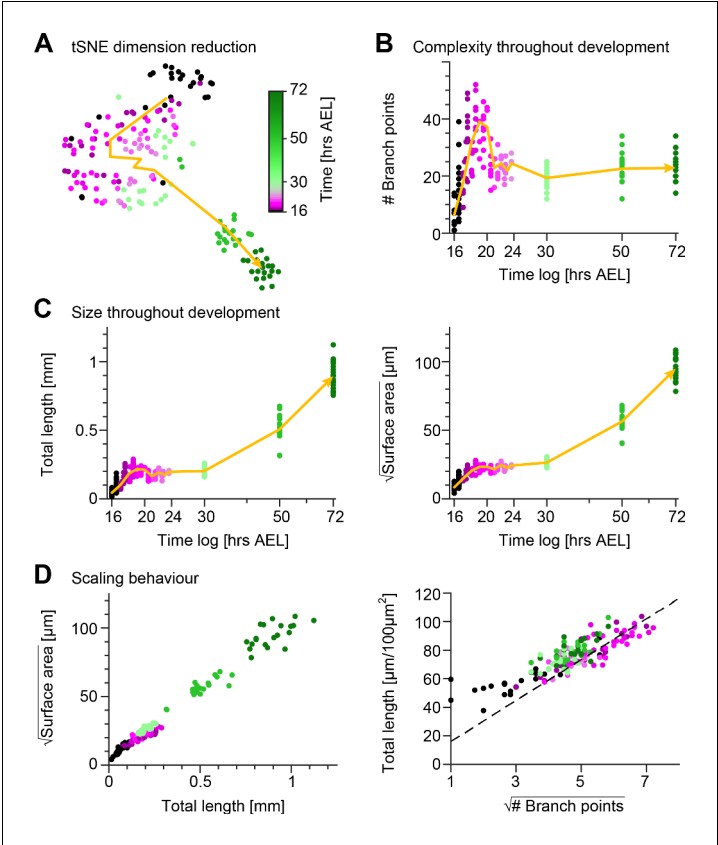

**Figure 2.** Quantification of c1vpda dendrite differentiation throughout development. (**A**) A t-distributed stochastic neighbour embedding (tSNE) plot showing the entire dataset of neuronal reconstructions using a 49—dimensional morphometric characterisation reduced to two dimensions. (**B**) Time course of the number of branch points during development (see also *Figure 6C*). (**C**) Time courses of the total length of dendrite cable (left) and square root of the surface area (right) during development (see also *Figure 6C*). (**D**) Scaling behaviour of the square root of the surface area against total length (left) and total length against number of branch points (right) showing the relationships expected from the optimal wire equations (*Cuntz et al., 2012*; *Baltruschat et al., 2020*). The dashed line shows the average scaling behaviour of simulated synthetic trees ($n = 1,000$ simulations; see Materials and methods). In all panels, each dot represents one reconstruction with the colour scheme indicating imaging time AEL roughly dissecting embryonic (magenta) and larval (green) developmental stages (colour bar in A). The thick yellow arrows show trajectories averaging values of all reconstructions across two hour bins in A, and 1 hr bins in B and C for higher resolution. Data from $n = 165$ reconstructions, $n = 48$ neurons, $n = 13$ animals. See also *Figure 2—figure supplement 1* for details on the scaling in the different stages of development.

The online version of this article includes the following figure supplement(s) for figure 2:

**Figure supplement 1.** Scaling relations between key morphometrics during development.

*Baltruschat et al., 2020*). For planar dendrites that minimise wire, a scaling law relating branch points (*N*), total length (*L*) and surface area of the spanning field (*S*) was formerly derived (*Cuntz et al., 2012*):

$$L \approx \sqrt{\frac{1}{\pi}} \cdot \sqrt{S} \cdot \sqrt{N}. \tag{1}$$

Thus, as a first step to assess if c1vpda sensory neurons saved wire during development we verified if their dendrites obeyed the expected geometrical square root scaling relationship. As predicted by the aforementioned equation, a square root relation between dendrite length *L* and surface area *S*, and a square root relation between total length *L* and number of branch points *N* were found at each developmental time point (*Figure 2D*; see Materials and methods). In the scaling plot of the length *L* and surface area *S*, the slight offset between the light green and magenta dots

marks the stage transition between embryonic growth and the subsequent isometric stretching observed during instar stages.

To further test the wire minimisation properties of c1vpda neurons we compared the scaling relations of synthetic dendritic morphologies against real data (see Materials and methods). Synthetic trees were generated using a formerly described minimum spanning tree (MST) based model and were simulated to match the morphometrics of the real neurons (*Cuntz et al., 2008*; *Cuntz et al., 2010*). Note that this model does not capture the evolution of the developmental process, and it is only designed to generate morphologies that replicate the shape of real cells at static developmental time points. To facilitate comparing the total length and number of branch points of the datasets, artificial and real morphologies were normalised to a standard arbitrary surface area of 100 μm². As a result, we could then show that the square root of the number of branch points $\sqrt{N}$ and total length $L$ of the synthetic trees scaled linearly with each other, with the experimental data being well fitted by the synthetic data ($R^2 = 0.98$, *Figure 2D*).

Taken together, the results indicate that throughout morphological differentiation during development, c1vpda sensory neurons respect minimum wire constraints. This suggests that while functional requirements for dendritic morphology here may shape the dendrites, these must also respect wire optimisation constraints.

## Embryonic phase of branch retraction leads to c1vpda comb-like shape

Having established that the specification of c1vpda dendrite patterning essentially occurs during embryonic stages, we focused on how the embryonic retraction phase reorganises the tree structure. The time series of c1vpda growth in *Figure 1A* suggested that smaller, dorsoventral oriented, higher-order lateral branches were preferentially eliminated in the embryonic retraction phase, leaving most second-order lateral branches intact. This is interesting, as the innervation of the antero-posterior axis by post-embryonic c1vpda second-order lateral branches may play a role in sampling cuticle folding during crawling behaviour (*Vaadia et al., 2019*).

We therefore investigated the effects of the retraction phase on the spatial distribution of lateral branches, measuring their orientation before and after retraction. Imaging the immobile embryo did not enable us to directly measure the branch orientation of the imaged cells in relation to the direction of the body wall contraction during crawling. Therefore, we took advantage of the stereotypical c1vpda structure and location in the body of the larva and defined the MB as perpendicular to the direction of contraction. We then measured the angle of a given lateral branch in relation to the MB as a proxy for the direction of contraction (*Figure 3—figure supplement 1*, see Materials and methods). The orientation angle varied between 90° for a lateral branch aligned along the anteroposterior axis, e.g. some second-order lateral branches, to 0° for a branch extending in the dorsoventral axis, e.g. the MB (*Figure 3A*). The angles were measured separately in the longer second-order lateral branches emanating directly from the MB (order 2, blue branches) and in higher-order lateral branches which branch out from the second-order lateral branches (order > 2, orange branches).

Before the actual retraction phase (*Figure 3A*), that is, at the peak of branching complexity, higher-order lateral branches were shorter (with a median of 1.6 μm) and exhibited lower median angles (37.31°), than second-order lateral branches (6 μm, 64.72°, respectively, $p<0.001$, $p<0.001$ by bootstrap). Interestingly, the median branch lengths and angles of second-order lateral branches (6.1 μm and 63.93°) and higher-order lateral branches (2 μm and 41.67°) remained similar after retraction (*Figure 3B*). However, a drastic reduction in the overall number of branches was asymmetrically distributed between the different branch orders. The reduction of higher-order lateral branches (267 branches before retraction vs. 92 branches after retraction, with a decrease of −64.9%) greatly exceeded the reduction of second-order lateral branches (162 branches before retraction vs. 131 branches after retraction, with a decrease of −19.1%).

Importantly, the retraction stage seemed to reshape the overall branch angle distributions towards higher angles, that is, further oriented along the anteroposterior axis (*Figure 3*). Through the reduction of the higher-order lateral branches with their flat angle distributions the contribution of the peak at higher angles from the second-order lateral branches became more prominent (with an overall median angle pre retraction of 49.41°, and an overall median angle post retraction of 59.4°, a difference of the median of 9.99°, $p<0.01$, by bootstrap).

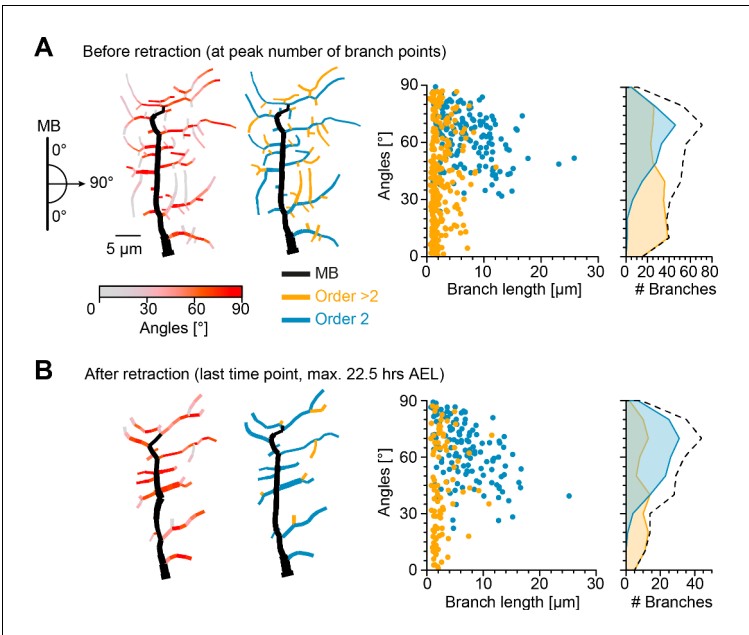

**Figure 3.** Retraction phase preferentially targets smaller, low orientation angle, higher-order lateral branches. (**A**) Sketch illustrating lateral branch orientation angle and dendrite morphology of a sample c1vpda sensory neuron before retraction. Morphology on the left side is colour coded by branch segment angles and morphology on the right is colour coded by branch length order (MB is coloured in black; see Materials and methods). On the right, histograms for branch length (one dot per branch) and number of branches per angle are shown separated by branch length order (blue: order 2, orange: order > 2, $n = 429$ branches). Dashed line represents the overall distribution of number of branches per angle. (**B**) Similar visualisation but for dendrites after retraction ($n = 223$ branches).

The online version of this article includes the following figure supplement(s) for figure 3:

**Figure supplement 1.** Illustration of the algorithm identifying the MB.

## C1vpda dendrites may facilitate mechanosensory signal transduction

The unbalanced retraction of higher-order lateral branches leading to a more anteroposterior oriented and comb-like morphology most likely has functional consequences. A recent study proposed that the integration of mechanical cues by c1da sensory neurons through activation of mechanogated ion channels depends on the curvature of individual dendritic branches (*He et al., 2019*). However, it remains unclear whether c1da dendritic branches are spatially arranged to maximise mechanical cue transduction through curvature.

Inspired by the results from *Vaadia et al., 2019* that demonstrated somatic c1vpda Ca²⁺ activation, we measured dendritic Ca²⁺ responses in freely forward moving larvae following branch deformation due to body wall contraction (*Figure 4A*, see Materials and methods and *Figure 4—figure supplement 1*). We generated a fly line in which c1vpda neurons specifically express tdTomato (red) as a fluorescent marker to visualise the dendrites and at the same time also GCaMP6m (green) to report changes in cytoplasmic Ca²⁺ concentration in the neuron. Thus, we measured segment contraction as an indicator of branch curvature and calculated the overall calcium signal transient of all apical branches of a given neuron. The mean GCaMP fluorescence peak ($\frac{\Delta R}{R_0}$) appeared with a short lag of 0.2 s after the maximum segment contraction, the actual limit of the temporal resolution of the set-up. Moreover, the GCaMP signal and the segment contraction correlated very strongly ($r = 0.85$, $p<0.001$, by Pearson coefficient). Ca²⁺ signals decreased as the peristaltic wave advanced to adjacent anterior segments (see *Figure 4—videos 1–3*). These data replicate the results previously found by *Vaadia et al., 2019*, supporting the finding that c1vpda sensory neurons respond to body wall folding during segment contraction with prominent Ca²⁺ signals in the dendrites.

We then modelled c1vpda membrane curvature, to simulate the effects of morphological alterations in the lateral branches due to cuticle folding during segment contraction. We designed a

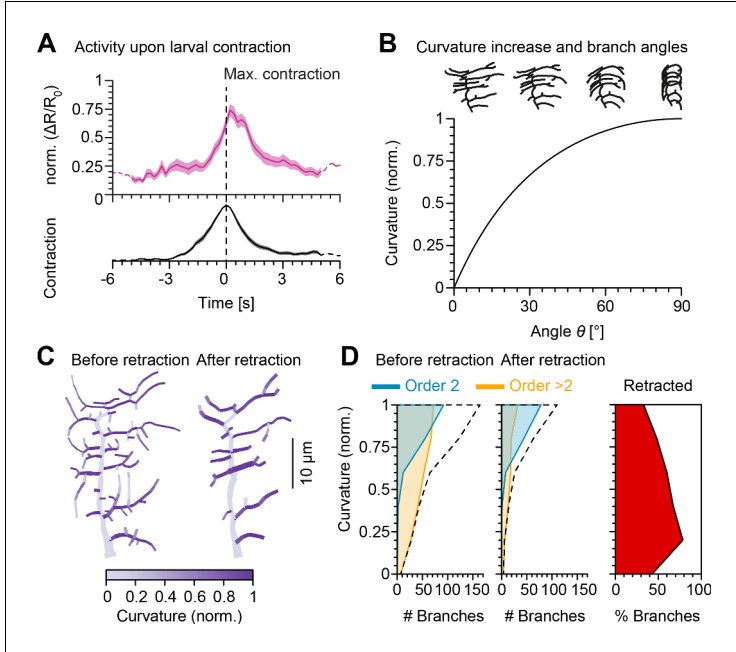

**Figure 4.** Retraction increases branch bending curvature during larval contraction potentially facilitating signal transduction. (**A**), (Top) Mean normalised Ca$^{2+}$ responses of c1vpda dendrites during forward crawling. The signal was calculated as the fold change of the signal $R = \frac{F_{GCaMP6m}}{F_{tdTomato}}$ in the fluorescence ratio $\frac{\Delta R}{R_0}$ (see Materials and methods and **Figure 4—figure supplement 1**). $\frac{\Delta R}{R_0}$ signal amplitude was normalised for each trial. Data from six animals, $n = 25$ neurons; solid pink line shows average values where data comes from $n>5$ neurons and dashed pink line where $n<5$ neurons. Standard error of the mean in pink shaded area. (Bottom) Average normalised contraction rate during crawling behaviour (similar plot as in top panel but in black colour). Segment contraction and Ca$^{2+}$ responses were aligned to maximal segment contraction at $t = 0s$. (**B**) (Top row) Simulated contraction of a c1vpda morphology by wrapping around a cylinder. (Main panel) Relationship between normalised curvature increase experienced by a single branch as a function of its orientation angle θ. (**C**) C1vpda dendrite morphologies before (left) and after (right) retraction. Morphologies are colour coded by local curvature increase during segment contraction. (**D**) Similar visualisation of the same data as in **Figure 3** but for curvature increase before and after retraction. Dashed line represents the overall distribution of number of branches per curvature increase. Rightmost panel additionally shows the distribution (%) of retracted branches by bending curvature increase (red shaded area).

The online version of this article includes the following video and figure supplement(s) for figure 4:

**Figure supplement 1.** Calcium imaging of c1vpda dendrites during forward crawling.

**Figure supplement 2.** Tubular structure elliptical profile approximation.

**Figure 4—video 1.** Video of c1vpda dendrites during forward crawling in real-time.
https://elifesciences.org/articles/60920#fig4video1

**Figure 4—video 2.** Video of c1vpda dendrites during forward crawling in real-time.
https://elifesciences.org/articles/60920#fig4video2

**Figure 4—video 3.** Video of c1vpda dendrites during forward crawling in real-time.
https://elifesciences.org/articles/60920#fig4video3

geometrical model of tubular structure bending, to measure the relative curvature increase of a given branch from resting state to the point of maximum segment contraction in relation with its orientation (see Materials and methods; **Figure 4—figure supplement 2**). The orientation angle of the tubes representing dendrite branches varied from $0° \leq \theta \leq 90°$ with respect to the direction of contraction ($\theta = 0°$ perpendicular; $\theta = 90°$ parallel to the direction of contraction). We then plotted the normalised branch curvature increase as a function of the orientation angle. As shown in **Figure 4B**, branch curvature increased steadily with the increase of the respective orientation angle independently of branch length or the size of the cylinder. Our data and modelling indicate that dendritic

branches extending along the anteroposterior body axis may be in the optimal orientation for bending during segment contraction (*Figure 4C*).

To explore this further in the context of retraction, we computed the relative bending curvature of lateral branches in c1vpda morphologies before (median of 0.93 for second-order lateral and of 0.71 for higher-order lateral branches with a difference between medians of 0.22, $p<0.001$, by bootstrap) and after retraction (median of 0.93 for second-order lateral and of 0.76 for higher-order lateral branches with a difference between medians of 0.18, $p<0.001$, by bootstrap) (*Figure 4D*). Similarly to the angle orientation measured in *Figure 3*, the retraction of predominantly higher-order lateral branches led to an overall higher median bending curvature (7.6% increase, $p<0.001$, by bootstrap). The increment was caused by the retraction of low bending curvature branches (*Figure 4D*). Taken together, these data and simulations suggest that functional constraints of mechanical responsiveness may represent a strong determinant in c1vpda dendrites patterning.

## In-silico simulations and in vivo branch dynamics are consistent with a stochastic retraction

Having established a putative functional role of the retraction phase, it is interesting to determine the precise principles upon which branch retraction operates. Is a selective retraction of higher-order lateral branches, or one that is specific to branches with non-optimal angles most consistent with the data at hand? To address this question we simulated *in silico* a variety of extreme schemes that selectively retract specific types of lateral branches from real morphologies (see Materials and methods). We computed the difference in number of branches between individual c1vpda neurons before and after retraction to then simulate the morphological effects of removing the same amount of branches on those morphologies using different retraction schemes. For each simulation, all branches were sorted according to their morphology, including length, orientation angle and branch length order (BLO). Afterwards, branches were selected to be retracted as specified in the following conditions: (1) Short branches first; (2) Branches with low angles first; (3) Lateral branches with higher branch length order first; Finally, (4) a stochastic retraction process as a control. These retraction schemes were each applied on the real morphologies at the time point exactly before retraction initiated until the post-retraction number of branches was reached. The resulting simulated trees were then compared with the real morphology after retraction (*Figure 5—figure supplement 1*). Surprisingly, the random retraction was the only scheme that yielded good results across all morphometrics compared to the experimental data.

Our simulations exclude the more extreme versions of some possible retraction schemes while identifying a random retraction of branches as a potential candidate to explain the biological data. This type of random retraction could be responsible for developing the c1vpda comb-like shape in a self-organised manner that may be less costly to genetically encode than a deterministic retraction program (*Hiesinger and Hassan, 2018*). Interestingly, this would make the random retraction scheme efficient at realising functionally specialised morphologies while being itself potentially the product of a rather non-specialised genetic program. In order to better understand the dynamics of this process and its interactions with branch outgrowth we performed time-lapse analysis at the single branch resolution (see Materials and methods). For this analysis, branches were classified into one of the following five types: retracted, shortened, new, elongated, and stable branches (*Figure 5A*).

Interestingly, when measuring the rates of extension and reduction by tracking individual lateral branches, we found that all types of branches maintained a moderately constant trend throughout the retraction phase (*Figure 5B*). Both reduction and extension averaged approximately between 2 and $3 \frac{\mu m}{hr}$ in all cases. This analysis suggests a branch type and time invariant mechanism of branch extension and reduction in c1vpda sensory neurons. Note that the results are the net value of branch tip position between observation points at time intervals of 1 hr. Even though the interpretation of this analysis is valid for the selected time interval, higher temporal resolution will no doubt uncover higher frequency dynamics that are not captured here, a common problem of any time-lapse analysis and indeed of any type of image processing (*Helmstaedter, 2013*; *Peng et al., 2017*). However, the net changes within 1 hr intervals are a stable result and provide for a phenomenological description and quantification of the branch reorganisation throughout the retraction phase.

Since the rates of extension and reduction were similar throughout, the specific proportion of branches per branch type must vary across the examined development window in order to

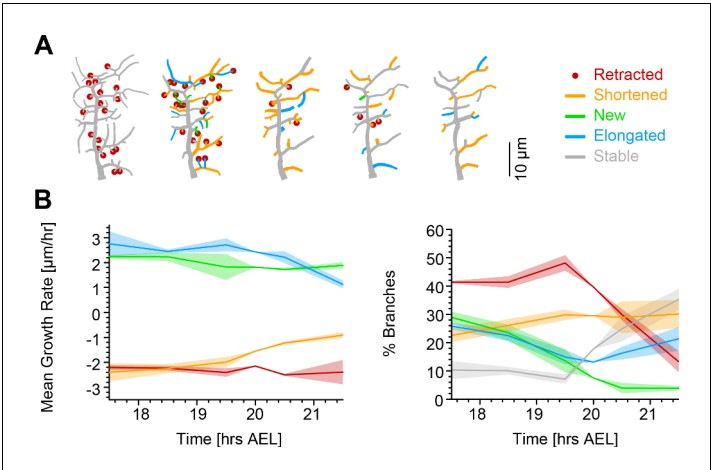

**Figure 5.** Single branch tracking analysis quantifies retraction phase dynamics. (**A**) Dynamics of retraction phase for one sample c1vpda dendritic morphology with branches coloured by their respective dynamics, red circles–to be retracted; orange–shortened; green–newly formed; blue–elongating; grey–stable. (**B**) (Left) Branch dynamics similar to A but quantified as growth rates ($\frac{\mu m}{hr}$) for all branches of all dendrites tracked during the retraction phase, $n = 1,139$; same colours as in A. (Right) Assignment of branches to the five types in A as a function of time. Shading represents the standard error of the mean.

The online version of this article includes the following figure supplement(s) for figure 5:

**Figure supplement 1.** *In silico* simulations to quantify retraction phase dynamics.

accommodate a retraction phase. Indeed, an initial phase of more intense branch dynamics, with only a small amount of branches remaining stable, lasted approximately half of the analysed time period. In that period of time, roughly half of the branches were involved in retraction while the number of new and elongating branches decreased steadily over time (*Figure 5B*). This was followed by a phase defined by the sharp decrease in the number of retracting branches, contrasting with the increase of stable branches, corresponding to the initiation of the stabilisation stage. In this latter phase, the number of new branches kept decreasing to virtually negligible values. In the same time, the proportion of elongating branches increased back to efficiently compensate for the remaining shortening further contributing to the stabilisation phase. In conclusion, both our retraction simulations as well as measurements of single branch dynamics indicate that retraction is neither specific to functionally suboptimal branches, nor to smaller or higher-order branches but stochastic in nature. Nevertheless, the stochasticity of retraction does not prevent it from supporting optimal mechanical responsiveness as shown above.

## Computational growth model reproduces c1vpda dendrite development

In order to better understand how the retraction phase improves c1vpda branch orientation and how it complements the outgrowth phase to produce functionally efficient dendritic patterns, we designed a computational model simulating c1vpda development based on the time-lapse data. The model was based on previous morphological models that satisfy optimal wire considerations through minimising total dendritic cable and conduction times from dendrite tips to the soma (*Cuntz et al., 2007*; *Cuntz et al., 2010*; *Cuntz et al., 2008*). It differs from this MST based model by capturing spatiotemporal differentiation of dendritogenesis in a continuous manner. In particular, it relied on a recent model designed for class IV da (c4da) neurons that satisfies wire constraints while reproducing the iterations of dendrite growth during development (*Baltruschat et al., 2020*).

The c1vpda growth model reproduces the patterning of real neurons by simulating branch dynamics on a synthetic dendritic tree at a given time point to produce the tree in the following time point. The c1vpda model was constructed on a set of iterative local rules which represent dendrite branch growth of c1vpda sensory neurons, involving only three processes: branch elongation,

interstitial branching and branch retraction. The numerical simulations were performed within the 2D physical boundaries of the spanning area of real neurons (see Materials and methods).

Synthetic growth started with the polarisation (P) of the MB. Then, lateral branch morphogenesis initiated with second-order lateral branches sprouting from the MB, and higher-order lateral branches emerging from those branches (*Figure 6A*). New branches and elongating branches grew away from existing synthetic dendrites in the direction of target points, while remaining within a given growth radius defined as the average length of newly formed branches quantified in the single branch tracking analysis. The target points were stochastically selected from within the spanning area of the cell. In parallel, other branches were randomly selected to be shortened, and they were retracted in case their length was equal or less than the retraction length defined as the mean length of retracted branches found in the single branch tracking analysis. The distribution of new and retracting branches over time were obtained directly from the time-lapse data in *Figure 2B* and *Figure 5B* without recurring to any parameter fitting (see Materials and methods).

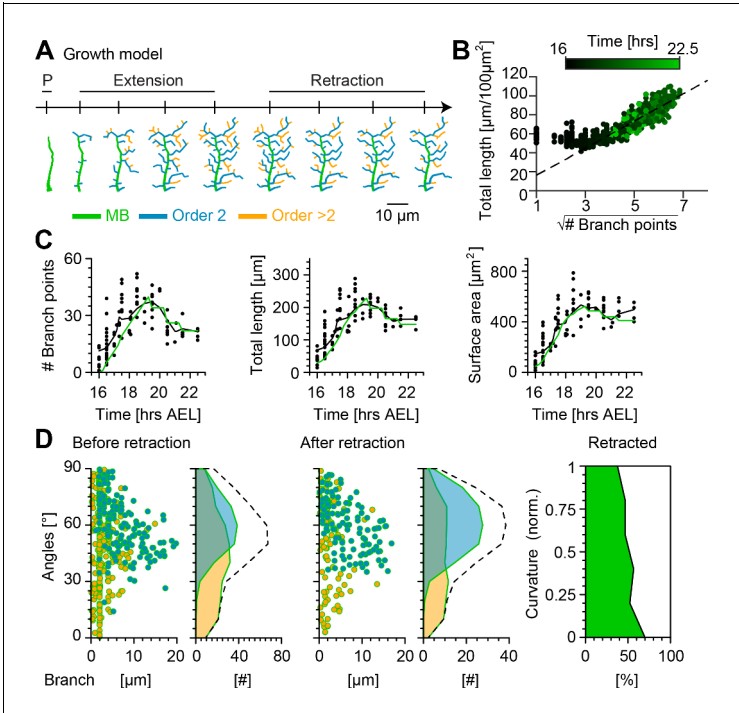

**Figure 6.** Computational growth model with stochastic retraction satisfies optimal wire constraints and replicates c1vpda dendrite growth. (A) Synthetic dendrite morphologies of a sample c1vpda during the entire embryonic development ordered by their respective developmental stages: polarisation (P), extension and retraction, until the stabilisation phase. (B) Scaling behaviour of total length against the square root of branch points of the trees generated using the random retraction growth model. The dashed line shows the average scaling behaviour of the simulated MST trees used in *Figure 2D* ($n = 1,000$ simulations; $R^2 = 0.98$; see Materials and methods). (C) Time course of the number of branch points ($R^2 = 0.88$), total length of dendrite cable ($R^2 = 0.95$) and surface area ($R^2 = 0.94$) during development until the stabilisation phase. In all panels, each black dot represents one reconstruction ($n = 90$) black solid lines represent the moving average of the real neurons and green solid lines represent the mean behaviour of the synthetic trees ($n = 1,215$). (D) Representative visualisation of a random sample of synthetic trees before retraction (left, with same number of trees as in experimental data) histograms for branch length (one dot per branch) and number of branches per angle are shown separated by branch length order (Blue: order 2, Orange: order > 2). Dashed line represents the overall distribution of number of branches per angle. Similar visualisation (middle) of dendrites after retraction as well as summary histograms. Rightmost panel shows the distribution (%) of retracted branches by bending curvature increase (green shaded area).

The online version of this article includes the following figure supplement(s) for figure 6:

**Figure supplement 1.** Growth model without retraction does not replicate c1vpda dendrite growth.

The number of branch points, total length and surface area were consistently well fitted by the growth model with random retraction at all simulated developmental stages (*Figure 6C* and *Figure 6—figure supplement 1*). Importantly, the model reproduced the scaling relationships from *Figure 2D*, indicating that the morphologies resulting from the random retraction growth model followed basic wire constraints as observed in the real data (*Figure 6B*). The results also showed remarkably good correspondence with other key morphometrics.

The model strengthened the hypothesis that a stochastic retraction was responsible for arbour refinement in c1vpda sensory neurons. The model branch length and angle distributions before and after retraction matched the real data (*Figure 6D*, *Figure 3*) as well as the selective retraction of lower curvature branches observed in *Figure 4* (lower curvature branches ranging between 0−0.2 retraction = −70% vs. higher curvature branches ranging between 1−0.8 retraction = −37.64%; *Figure 6D*).

All together, these findings indicate that a stochastic growth that satisfies wire constraints combined with random retraction of terminals are consistent with c1vpda dendrite morphogenesis and refinement. However, although the random retraction model successfully reproduced the most significant morphometrics of the experimental data, we note that the lateral branches from the model slightly under-estimated the orientation angle of the second-order lateral branches after retraction (median model = 60.18° vs. median real neurons = 63.93°). This indicates that possibly other mechanisms may be involved in enhancing tips growth direction preference, such as specific cell adhesion molecules (*Hattori et al., 2013*).

## Discussion

We have shown that the spatiotemporal patterning of c1vpda mechanosensory dendrites during development can be accurately predicted by a noisy growth model that conserves wire, in combination with a stochastic retraction that plausibly enhances their performance at sensing larval contractions. Using single branch tracking analysis on long-term time-lapse reconstructions, we were able to constrain the model without recurring to parameter fitting. We showed how a sequence of three simple stages (1) MB polarisation, (2) subsequent branch outgrowth and (3) a final stochastic retraction stage generates specialised dendrites that favour functional branches, as found in real c1vpda sensory neurons.

### A noisy growth process underlies morphological differentiation

C1vpda development started with the polarisation of the MB. The growth direction of the MB was constant across cells, with the MB of neurons from different hemisegments projecting dorsally, parallel to each other (see *Figure 1*). During the subsequent extension phase, newly formed lateral branches emerged interstitially from the existing MB. These observations raise an interesting topic for future studies underlying the role of the direction of polarisation of the primary branches in positioning subsequent newly formed branches in the dendritic field (*Yoong et al., 2019*). Prior work in neuronal circuit wiring showed how a multistage developmental program that incorporates stochastic processes can generate stereotypical phenotypical outcomes (*Langen et al., 2015*). This counter-intuitive phenomenon is made possible in part by molecular mechanisms that utilise stochasticity to implement simple patterning rules (*Hiesinger and Hassan, 2018*; *Hassan and Hiesinger, 2015*; *Johnston and Desplan, 2010*; *Courgeon and Desplan, 2019*). Arguably for the case of c1vpda neurons, a combination of MB orientation, noisy filopodial exploration and contact-based local decisions on where to grow using *Dscam* based self-avoidance synergised to coordinate lateral branch patterning (*Grueber et al., 2003a*; *Zipursky and Grueber, 2013*; *Matthews et al., 2007*; *Hughes and Thomas, 2007*; *Soba et al., 2007*; *Dong et al., 2015*).

The initial innervation of the dendrite's spanning field by lateral branches produced – similarly to previously observed class IV da neurons – optimally wired (see *Figure 2*) and space-filling dendrites (*Baltruschat et al., 2020*). In the c1vpda the branches at this early stage divided into two distinct morphological classes: (1) longer second-order lateral branches that spread along the anteroposterior axis with growing tips mostly targeting distal and sparser areas of the dendritic territory. (2) In contrast, higher-order lateral branches exhibited shorter lengths, mainly innervating the dorsoventral axis, and more often located in proximal and densely packed areas of the dendrite's spanning field.

## Phases of c1vpda development

Following the extension phase, we observed a retraction step that refined the spatial arrangement of the dendritic tree (*Figure 3*). In the past, studies based on low temporal resolution static data of dendrite development suggested that distinct growth and retraction phases may happen sequentially during development (*Lázár, 1973*; *Huttenlocher and Dabholkar, 1997*; *Bystron et al., 2008*). At higher temporal resolution, time-lapse movies showed that branch additions and retractions seem to happen rather concurrently during arbour elaboration (*Hua and Smith, 2004*; *Hossain et al., 2012*; *Cline, 2001*; *Dailey and Smith, 1996*; *Yoong et al., 2019*). In our dataset on c1vpda, we observed parallel growth and retraction of branches with changes in their proportions leading to separate phases of predominant outgrowth, retraction and then stabilisation.

The retraction of a dendritic tree could have economical purposes and minimise the amount of wire, or it could refine the branching pattern to enhance functionality. Our data indicate that the latter is the case, with a simple random retraction selectively remodelling the tree structure, influencing the mechanisms of dendritic signal integration (*Figure 5—figure supplement 1*). This result was surprising at first because it suggested that to ensure the removal of sub-optimal branches retraction effectors could be spatially constrained around higher-order lateral branches or branches with low orientation angle, exerting control over their elimination. However, the biased retraction of higher-order lateral branches was really attained due to the combination of three factors: asymmetry of branch length distributions between branch orders (*Figure 3*), branch reduction and extension rates similarity and invariance in time and across branch orders, and the increase of the proportion of branch reductions during the retraction phase (*Figure 5B*). Taken together, the random selection of a large number of branches to retract a constant amount of cable from their tips led to the penalisation of higher-order lateral branches due to their smaller lengths. In contrast, second-order lateral branches characterised by longer lengths retracted less.

An interesting question that arises from the present study is: what is the mechanism that generates the stochastic retraction observed in c1vpda neurons? We conceive distinct possibilities that are not mutually exclusive. The first possibility is that a genetically determined mechanism would cause the observed stochastic retraction at a stereotypical developmental time. We speculate as well that the retraction period could be activity-dependent. It has been demonstrated that activity coordinates the development of functional neurons in visual and motor systems of *Drosophila* (*Fushiki et al., 2013*; *Akin et al., 2019*). For the case of c1vpda neurons, the observed peristaltic waves during embryonic stages could potentially activate these cells and signal the beginning of the retraction phase and subsequent cell maturation. Alternatively, as the number of dendrite branches and cable increase during the extension phase competition-based mechanisms could provide adaptive negative feedback on branch growth, to avoid uncontrolled innervation. These mechanisms may either result from different levels of intrinsic stabilisation in the different types of branches that could lead to the elimination of the smaller branches and weakly supported tips, or branches contacting in densely packed areas of the dendritic field could drive tips to mutually retract (*Palavalli et al., 2020*). The latter hypothesis potentially involving Dscam is of particular interest since we proposed previously that Dscam based molecular machinery may help implementing a developmental program computationally equivalent to minimum spanning tree algorithms (*Cuntz et al., 2010*; *Cuntz et al., 2012*).

After the retraction step, c1vpda trees went through a stabilisation period, characterised by a negligible increase in cable length, surface area, and number of branches through a net balance of shortening and elongation of branches (see *Figure 1*). After hatching, the dendrites experienced an isometric scaling, where the comb-like pattern and branching complexity of the dendrites persisted across all larval stages, and the cable and surface were increased following the larva's body growth (*Parrish et al., 2009*). The conservation of dendrite shape throughout larval stages suggests the need for functional conservation during larval development. This observation is consistent with previously reported results, that showed that the behavioural repertoire of L1 larvae was analogous to L3 larvae (*Almeida-Carvalho et al., 2017*). However, besides fulfilling their functional role, c1vpda neurons also optimise resources. The overproduction of material carries a fitness cost to the organism and as a result a trade-off between function and resources conservation arises (*Szekely et al., 2013*; *Wen and Chklovskii, 2008*; *Cuntz et al., 2007*; *Bullmore and Sporns, 2012*). This trade-off between function implementation and wire optimisation in dendrites raised the possibility that to

implement this important function more cable could have been spent to achieve a highly specialised pattern in detriment of wire minimisation. Here, we showed that a noisy growth process with different stages optimises function, structure and wire in a self-organised manner.

## An improved computational morphological model

We developed computational growth models that included stochastic retraction in c1vpda dendrites (*Figure 6*). In the past, a variety of models have been proposed to generate neuronal morphologies reproducing morphometrics of real mature dendrites (*Cuntz et al., 2007*; *Cuntz et al., 2010*; *Donohue and Ascoli, 2008*; *Eberhard et al., 2006*; *Koene et al., 2009*; *Torben-Nielsen and De Schutter, 2014*; *Luczak, 2006*; *Beining et al., 2017*). However, some of these growth models rely on large number of parameters that are not available from experimental data, and they tend to provide phenomenological insights rather than a mechanistic understanding of a given system (*Goodhill, 2018*). Most notably, none of those approaches have specifically modelled development quantitatively (but see *Yalgin et al., 2015*; *Sugimura et al., 2007*; *Baltruschat et al., 2020*). Also none of those approaches have focused on a quantitative understanding of retraction in the developmental process even though the importance has been emphasised widely (*van Pelt, 1997*; *Beining et al., 2017*; *Luczak, 2006*; *Torben-Nielsen and De Schutter, 2014*; *Williams and Truman, 2004*).

Having quantified the dynamics throughout development of both the growth and retraction of branches using high resolution time-lapse imaging, we were able to use these data to parameterise our model. Due to the lack of high-throughput automatic digital tracing algorithms for neuronal morphologies the chosen temporal resolution for imaging (see Materials and methods) was determined as a trade-off between capturing c1vpda dendrite dynamics throughout embryonic and larval stages and tracing speed (*Helmstaedter, 2013*). Branch dynamics and morphometrics at this stage were well defined by a stochastic growth and retraction model, suggesting that c1vpda morphogenesis is possibly a non-deterministic process, in accordance with other previously found results for other cell types (*Ryglewski et al., 2017*; *Özel et al., 2015*). The accordance between data and model further validates the chosen temporal resolution of imaging. In the future, as more sophisticated tracing algorithms develop, more detailed datasets will become available for analysis and modelling (*Peng et al., 2017*).

## Consequences for computation in dendrites

It was previously suggested that the stereotypical comb-like shaped c1vpda dendrites optimally sense the mechanical strain due to the hinge-like dynamics during cuticle folding (*Vaadia et al., 2019*; *He et al., 2019*). Interestingly, previous theoretical results on elastic properties of lipid bilayers showed that curvature is dependent on the orientation of the membrane (*Helfrich, 1973*; *Bahrami et al., 2016*). Based on theoretical predictions (*Figure 4*), we propose that the second-order lateral branches are better suited for mechanical sensory cues transduction arising from cuticle folding during crawling behaviour than higher-order lateral branches. Due to their direction preference running along the anteroposterior axis these branches experience larger curvature increase, possibly increasing the opening probability of the mechanogated ion channels (*Liang and Howard, 2018*; *Katta et al., 2015*; *Jin et al., 2020*). These results strengthen a recently proposed hypothesis, which predicted that similar sensory neurons (dorsal c1da mechanosensory neurons c1ddaE and c1ddaD) may become activated by membrane curvature increase (*He et al., 2019*).

Several findings are consistent with this hypothesis. Unique structural adaptations in the microtubule mesh of c1da sensory neurons support their role in sensing and responding to mechanical stimuli arising from the contraction of the body wall. C1da neurons contain denser arrays of microtubules in their branches than other da classes, and are firmly anchored to the epithelium by pads of electron dense material (*Delandre et al., 2016*). These structural adaptations are also present in other cells active in mechanotransduction (*Krieg et al., 2014*; *Liang et al., 2014*). Moreover, similar results were reported in *C. elegans*, suggesting that dendrite curvature may provide the biophysical substrate of mechanosensory experience across multiple animal models (*Albeg et al., 2011*; *Hall and Treinin, 2011*).

## Conclusions

Taken together, our results demonstrate that a specialised dendritic tree pattern that minimises wire can be obtained by the precise temporal arrangement of stochastic developmental programs. Interestingly, evidence can be found that similar stages and strategies may be preserved across different cell types (*Richardson and Shen, 2019*; *Gao et al., 2000*; *Sugimura et al., 2003*; *Baltruschat et al., 2020*) and species (*Yoong et al., 2019*). The flexible usage of such self-organisational programs provides developmental resilience and robustness to perturbations in the growth medium (*Hiesinger and Hassan, 2018*; *Johnston and Desplan, 2010*). It also possibly avoids the encoding of a deterministic morphogenetic program that may be more costly to implement genetically (*Hiesinger and Hassan, 2018*). In the future, it will be interesting to elucidate the mechanisms that control the temporal sequence of distinct stages of branch elaboration for the c1vpda sensory neurons (*Grueber et al., 2003a*; *Jinushi-Nakao et al., 2007*; *Nanda et al., 2019*) and on a higher scale to understand to what extent similar self-organising processes and mechanisms are implicated in the formation of other cell types (*Ryglewski et al., 2017*; *Özel et al., 2015*), neuronal networks (*Hassan and Hiesinger, 2015*) and even in the emergence of non-neuronal branching organs (*Hannezo et al., 2017*).

# Materials and methods

**Key resources table**

| Reagent type (species) or resource | Designation | Source or reference | Identifiers | Additional information |
|---|---|---|---|---|
| Genetic reagent (*D. melanogaster*) | *221* Gal4 | *Ye et al., 2004* | | |
| Genetic reagent (*D. melanogaster*) | *UAS-mCD8::GFP* | | RRID:BDSC_32187 | |
| Genetic reagent (*D. melanogaster*) | *UAS-IVS-GCaMP6m* | | RRID:BDSC_42748 | |
| Genetic reagent (*D. melanogaster*) | *UAS-CD4::tdTomato* | | RRID:BDSC_35837 | |
| software, algorithm | Trees Toolbox (MATLAB) | *Cuntz et al., 2010* | RRID:SCR_010457 (RRID:SCR_001622) | treestoolbox.org (mathworks.com) |
| software, algorithm | FIJI | *Schindelin et al., 2012* | RRID:SCR_002285 | fiji.sc |

### *Drosophila* lines

Flies were reared on standard food in a 12 hrs light-dark cycle at 25°C and 60% humidity unless otherwise indicated. For time-lapse visualisation of the dendritic tree structure of c1vpda sensory neurons in the embryo and at stages L1, L2 and L3 221-Gal4 (*Ye et al., 2004*) was recombined with *UAS-mCD8::GFP* (Bloomington stock #32187). For in vivo imaging of dendritic calcium dynamics and dendritic structure simultaneously, flies carried the c1vpda sensory neuron driver *221* Gal4, the calcium indicator *UAS-IVS-GCaMP6m* (Bloomington stock #42748) and the membrane marker *UAS-CD4::tdTomato* (Bloomington stock #35837).

### Time-lapse image acquisition

In the embryo (seven animals), 28 neurons were imaged at 5*mins* resolution between 16 hrs AEL and around 24 hrs AEL (*Figure 1A*), for periods ranging from 30*mins* to 6 hrs. Image stacks from the time series were reconstructed at 30*mins* and 1 hr intervals. Starting at around 22.5 hrs AEL light peristalsis waves were observed in the embryo, but the imaging sessions continued until around 24 hrs AEL. After hatching, 20 neurons (five animals) were imaged at time points 30 hrs AEL, 50 hrs AEL and 72 hrs AEL, to cover larval development. Mouth hooks and molting were used as developmental markers to define the correct time points to image c1vpda sensory neurons in L1, L2, and L3 (*Park et al., 2002*).

Images were acquired with a Zeiss LSM 780 Meta Confocal Microscope (https://www.zeiss.com). To keep the animals alive during the entire development, the laser intensity was kept to a minimum, especially in the early stages, to minimise the phototoxicity by exposure to the argon laser. In the embryo, to acquire high resolution images on the *z*-plane while minimising exposure to the argon laser, we decreased the imaging time per stack, by choosing a distance between the *z*-planes of 1 μm. For embryos, we used a 63 × 1.4 NA oil immersion objective and voxel size (0.2196 μm ×

0.2196 μm × 1 μm ) for seven time series, and for the remaining 21 time series we used a 40 × 1.4 NA oil immersion objective with voxel size (0.3459 μm × 0.3459 μm × 1 μm). During the L1 stage (30 hrs AEL), we used a 40 × 1.4 *NA* oil immersion and voxel sizes (0.4465 μm × 0.4465 μm × 1 μm) and (0.3907 μm × 0.3907 μm × 1 μm). When the image stacks using these voxel sizes were blurred we increased the resolution to (0.3907 μm × 0.3907 μm × 0.5635 μm). For L2 stages (50 hrs AEL), we used a 40 × 1.4 NA oil immersion objective and a wide range of voxel sizes – (0.5209 μm × 0.5209 μm × 1 μm), (0.4465 μm × 0.4465 μm × 1 μm), (0.3907 μm × 0.3907 μm × 1 μm) or (0.2841 μm × 0.2841 μm × 1 μm) to assure high resolution images for all cases. Finally, to acquire images during L3 stage (72 hrs AEL), we used a 20 × 0.8 NA multi-immersion objective and voxel sizes (0.8335 μm × 0.8335 μm × 1.5406 μm) and (0.7144 μm × 0.7144 μm × 1 μm).

## Embryo handling

Adult male and female flies were collected in a cage closed with an apple agar petri dish. Before embryo collection, a dab of yeast paste was added to a fresh apple agar plate. This first plate was removed and discarded after 1 hr and exchanged with a fresh plate with yeast paste. In this way, we assured that older and retained embryos were discarded. For the actual embryo collection, embryos were collected for 30*mins* and then allowed to age until the appropriate time for imaging. Until the imaging session started, the embryos were kept in the incubator at 25°C and 60% relative humidity on apple agar to prevent them from drying out.

Before the imaging session started, the embryos were dechorionated with mild bleach (50% Clorox; final concentration: 2.5% hypochlorite) for 3.5*mins*. Not all embryos were dechorionated by this gentle treatment, but only dechorionated embryos were selected to be imaged. After being selected, the embryos were handled using an artist's brush and were washed with water three times in a filtration apparatus.

## Embryo imaging

To immobilise the embryos to acquire well-aligned image stacks of the complete dendrite without damaging the egg, we designed a custom-made plate using *Autodesk Inventor 2016* (2019 Autodesk Inc) with dimensions of approximately (50 mm × 25 mm × 1 mm), with nine oval chambers carved on its surface with dimensions of approximately 3 mm × 1.2 mm × 0.2 mm. We printed the plate in white resin using a Form 2 (2019 Formlabs Inc) stereolithographic 3D printer. The embryos were deposited on the oval chambers and oriented in a way that the ventral side faced towards the cover slip. Halocarbon oil 700 (Sigma H8898) was deposited in the chambers to ensure oxygen access during imaging.

## Instar stages imaging

L1, L2 and L3 larvae were imaged under a custom-made chamber (*Dimitrova et al., 2008*) to curtail contact-based damage to the epidermis of the larvae. The chamber had three components: a metal plate, a plastic slide, and a round microstrainer that fitted a round cover slip. The larvae were positioned and immobilised between the cover slip and the microstrainer. The components were gently mounted with screws between the metal plate objective slide and the plastic slide. Again, throughout all imaging sessions the larvae were covered in halocarbon oil to ensure access to oxygen.

In between imaging sessions, every animal was kept at 25°C at 60% relative humidity in a separate 500 μl Eppendorf tube, which was filled with 200 μl flyfood. Holes were carved on the lid of the tube to guarantee air exchange. Before the next imaging trial, the flyfood was dissolved in water and the larvae were localised under a binocular microscope and washed three times with tap water.

## Functional imaging

Forward crawling imaging trials were performed in 25 neurons (A2–A6 segments) from 6 L1 larvae. Every imaging session lasted for 40 s. The imaging session was terminated and restarted when the larvae crawled entirely away from the field of view. The smaller body size at the L1 stage enabled a wide view of multiple ventral segments at the same time. The larvae were mounted on a glass slide with their ventral side facing the cover slip. The animals were imaged while immersed in Ringer solution (5 mM HEPES, pH 7.4, 130 mM NaCl, 5 mM KCl, 2 mM CaCl2, 2 mM MgCl2) in 1.1% low-melting agarose (TopVision Low Melting Point Agarose Thermo Fisher). The medium's high viscosity

caused resistance on the body of the larvae slowing down the crawling speed, enabling the acquisition of high resolution images of peristalsis.

Functional Calcium signals were acquired with a Zeiss LSM 780 Meta Confocal Microscope (https://www.zeiss.com). The imaging sessions were recorded in two different emission channels simultaneously, the green channel captured the *GCaMP6m* transients and the red channel captured dendrite deformation using membrane-tagged $CD4 - tomato$. Images were recorded at a temporal resolution of 0.2 s per frame, with 40 × 1.4 NA oil immersion objective with voxel size of $(1.3284 \mu m \times 1.3284 \mu m \times 1 \mu m)$.

## Contraction rate calculation

To quantify the body wall contraction rate, a triplet of adjacent c1vpda cell somata on the antero-posterior axis, were manually tracked during contraction–distension cycles of the crawling behaviour, using the *ImageJ* `Mtrack2` plug-in (*Meijering et al., 2012*) from *Fiji* (*Schindelin et al., 2012*). The contraction rate was calculated using *Matlab* (http://www.mathworks.com) as the sum of the Euclidean distances between the *x* and *y* coordinates of the central neuron and the *x* and *y* coordinates of the anterior and posterior neurons over time (*Figure 4—figure supplement 1*). In order to compare data across trials from different neurons and to avoid noise from different imaging sessions we normalised the contraction rates between the interval 0−1, where 0 corresponds to the maximum segment distention and 1 to the maximum segment contraction during peristalsis, that is, the minimum value of the sum of the Euclidean distances of a given triplet of neurons during a contraction–distension cycle.

## Dendrite region of interest (ROI)

The regions of interest (ROIs) in which to measure the $Ca^{2+}$ signal were first defined manually as a rough contour around the apical dendrite of the central cell of a given triplet for every time point of an imaging session, using the `ROI` (functionality from *Fiji*. Afterwards, we automatically generated tighter contours using the `'Defaultdark'` parameter from the `roiManager` menu (see available code)) by setting a threshold for the intensity values of the *tdTomato* signal, enabling the capture of pixels from the dendrite branches and not spurious noise in the larger ROI (*Figure 4—figure supplement 1*). Every ROI was defined on the red channel to capture dendrite cable tagged with $CD4 - tomato$, ensuring that the following $Ca^{2+}$ fluorescence extraction was done exactly on the c1vpda dendrite's membrane.

## $Ca^{2+}$ imaging analysis

The intensity values of *GCaMP6m* and *tdTomato* were then extracted for each ROI and time point and then exported from *Fiji*. The analysis of the fluorescence signals was performed using custom-made code in *Matlab* (http://www.mathworks.com). The *GCaMP6m* signal was normalised with the $CD4 - tomato$ signal and the ratio $R = \frac{F_{GCaMP6m}}{F_{tdTomato}}$ was used to calculate $\frac{\Delta R}{R_0}$. After the ratio between *GCaMP6m* and *tdTomato* was calculated, the background signal ($R_0$) was subtracted from every time point. $R_0$ was computed as the average of the first five frames of a given time series. Overall, the fold change of GCaMP6m fluorescence intensity over time was calculated as $\frac{\Delta R}{R_0} = \frac{R - R_0}{R_0}$. The function `unsharpmask` from *Fiji* (radius: 1.5, weight: 0.4) was applied to the images for visualisation in *Figure 4—figure supplement 1* to enhance dendrites, but the quantitative analysis was done with the raw imaging data.

To link $Ca^{2+}$ dynamics to the contraction of body wall experienced during crawling behaviour, we plotted the contraction rate against the $\frac{\Delta R}{R_0}$. However, as previously mentioned, the crawling speed can vary significantly between animals and across trials. Thus, to avoid averaging artefacts when comparing the $\frac{\Delta R}{R_0}$ transients against segment contraction, we first realigned the $Ca^{2+}$ traces to a biologically relevant marker. We chose to realign the $\frac{\Delta R}{R_0}$ according to the maximum segment contraction and only then calculated the mean of the signal.

## Modelling curvature increase

To understand how the bending of tubular membrane branches with different orientations affects their curvature, we assumed a marginal case for which the larva's cuticle folding can be

approximated by the surface of a cylinder with radius $R$ (*Figure 4—figure supplement 2A*). The orientation of the branch is then defined by the angle θ between the cylinder axis of symmetry and the central axis of each branch. The angle varies from $\theta = \frac{\pi}{2} = 90°$ for a branch oriented in the anteroposterior axis of the larva's body and perpendicular to the axis of symmetry of the cylinder in our model, to $\theta = 0 = 0°$ for a branch oriented in the dorsoventral axis of the larva's body and parallel to the axis of symmetry of the cylinder in our model. Starting from an initial branch with $\theta = \frac{\pi}{2}$ and length $L = 2\pi R$, we kept the branch length constant and calculated the curvature increase of the branch for different orientation angles $0 \leq \theta \leq \frac{\pi}{2}$. For simplicity we approximated the shape of a tilted branch, which follows an elliptical profile with diameters $a = R = \frac{L}{2\pi}$ and $b = \frac{a}{\sin\theta}$ on the cylinder, with a circular branch with a radius of curvature $R_c = 0.5(a + b)$ resulting in $\frac{1}{\sin\theta} = \frac{4\pi R_c}{L} - 1$ (see *Figure 4—figure supplement 2B*). An initial straight branch of radius $r$ has two principal curvatures $c_1 = 0$ and $c_2 = \frac{1}{r}$. Upon bending of the tubular branch around the cylindrical body with radius $R \gg r$, the second principal curvature is almost constant. Therefore, we computed the relative increase in the first principal curvature $c1$ to represent the curvature variation. The curvature increase is rescaled with respect to its maximal value for a branch oriented in the anteroposterior axis of the larva's body and perpendicular to the cylinder axis of symmetry with $\theta = \frac{\pi}{2}$. The curvature is a steadily rising function of the angle θ, varying from *zero* for a straight branch with $\theta = 0$ (see *Figure 4—figure supplement 2A*, bottom branch), to *one*, for a fully bent, that is, circular, branch with $\theta = \frac{\pi}{2}$ (see *Figure 4—figure supplement 2A*, left most branch; and *Figure 4B*).

## Dendrite morphometry

All morphometry analysis and stack reconstructions were performed in *Matlab* (http://www.mathworks.com) using our own software package, the *TREES Toolbox* (http://www.treestoolbox.org). Particularly, a number of new *TREES Toolbox* functions were custom-made and will be incorporated in the existing *TREES Toolbox* with publication of this work: `perpendicularity_c1_tree`, `turt_c1_tree`, `PB_c1_tree`, `features_c1_tree`, `BLO_c1_tree` and `isoneuronal_tree`. See below for details on the individual functions. In the following, typewriter typestyle function names with `_tree` suffix are *TREES Toolbox* functions.

## Stack reconstructions

Image stacks from the confocal microscope were imported in the TREES Toolbox environment and manual reconstructions of all apical dendrites were performed individually (N = 165) using the dedicated reconstruction user interface `cgui_tree`. During the reconstruction process, we determined adequate internode distances, that is, spatial resolution at which to resample (`resample_tree`) the dendritic structures, of 0.1 µm for smaller morphologies with total length smaller than 400 µm and of 1 µm for larger neurons with total length above 400 µm.

## Testing wire optimisation

To challenge the wire minimisation properties of c1vpda structure ($n = 165$) we verified if the branch points ($N$), total length ($L$) and surface area of the spanning field ($S$) obeyed the following scaling law $L \approx \pi^{-\frac{1}{2}} \cdot S^{\frac{1}{2}} \cdot N^{\frac{1}{2}}$ (*Cuntz et al., 2012*). The previously mentioned morphometrics were calculated using the functions `B_tree`, `len_tree` and `span_tree` respectively, from the TREES Toolbox (Matlab). Additionally, in order to further validate that c1vpda dendritic morphologies scale as expected by optimal wire principles, we implemented simplified models of dendritic trees based on the MST algorithm (`MST_tree`; bf = 0.2) (*Cuntz et al., 2010*). First, we generated MSTs to connect randomly distributed targets in a surface area of 100 µm². Targets were added until a maximum of 500 points as required to match the number of branch points of synthetic morphologies to the ones of real cells. These simulations were performed 1,000 times, totalling a number of $n = 500,000$ synthetic trees. In order to facilitate the comparison between total length and number of branch points of real dendritic trees from different developmental stages with the synthetic trees, all reconstructions were scaled to the same surface area (100 µm²) by using the function `scaleS_tree` (*TREES Toolbox*). Finally, the total length and number of branch points of the resulting real dendrites were compared with the ones from synthetic trees. The same procedure was then used to test the wire optimisation properties of the computational c1vpda growth model.

## Lateral branch orientation and curvature quantification

In order to compute the angle distribution and curvature increase of the lateral branches of c1vpda sensory neurons we wrote three custom *TREES Toolbox* functions: `PB_c1_tree`, `BLO_c1_tree` and `perpendicularity_c1_tree`. These functions are based in the following assumptions regarding the morphological properties of c1vpda dendrite structure and function observed in this study and in others before (*Grueber et al., 2003b*; *Vaadia et al., 2019*):

1. C1vpda sensory neurons show a topological bilateral symmetry. The MB that polarises from the soma is the central axis of symmetry that divides the lateral branches in the anterior and posterior directions.
2. Post-embryonic c1vpda sensory neurons (A1 – A6) are positioned in their corresponding segment with their MB dorsally oriented and running parallel to the MB of the adjacent posterior and anterior c1vpda neurons. Their lateral branches are oriented along the anteroposterior axis.
3. During crawling behaviour, peristaltic muscle contractions progress along the larva's body from posterior to anterior causing the lateral branches sprouting from the MB to bend due to cuticle folding, increasing their curvature. The MB remains virtually unmoved by the contraction motion.
4. The initial extension phase in the embryo generated a branched structure in both the anterior and posterior direction with the MB in the middle. This structure either produced long antero-posterior oriented paths between terminal nodes and the MB, or short dorsoventral oriented paths between terminals and MB (*Figure 1A*, middle row). The longer branches were reminiscent of the lateral branches in L1–L3 stages.

Taking into account assumptions 1−3, we measured the angles and curvature increase of the segments of a given dendrite branch in relation to the MB of the tree as a proxy for the direction of the body wall contraction. However, during development, the c1vpda sensory neurons migrate in the embryo changing their location and orientation relative to their initial position. This was also the case in larval instar stages where the dendritic orientation changed through different time points due to mechanical forces exerted on the larvae between the preparation and the cover slip during imaging sessions. We therefore required an unbiased procedure to reorient the dendrites.

To this effect we wrote the *TREES Toolbox* function `PB_c1_tree` that automatically finds the c1vpda MB and rotates the entire dendrite to align the MB to the y-axis (*Figure 3—figure supplement 1*). For a particular cell of interest the algorithm was initialised by finding the last node from the longest path (`pvec_tree` function) and rotating the tree (`rot_tree` function) until the last node was approximately aligned vertically (±1 μm) with the root at position (0, 0). This initialisation helped to reduce the number of computations required in the following steps of the algorithm (*Figure 3—figure supplement 1*).

Afterwards, a bounding box around the dendrite was computed using the `polyshape` and `boundingbox` functions (*Matlab*). The closest nodes of the tree to the top left and top right were then identified (`pdist`, *Matlab* function; see *Figure 3—figure supplement 1*). The first shared branch point between those two corners was then defined to be the last node of the MB (using `ipar_tree` function). Finally, the tree was rotated again until the new MB tip was approximately vertically aligned with the root at position (0, 0) (*Figure 3—figure supplement 1A*). The previous steps were repeated until no new last node was found between two consecutive iterations (*Figure 3—figure supplement 1B*).

After finding the MB of a given tree and taking into account assumption 3, we partitioned the tree into all the lateral subtrees that emerged from the MB. Each subtree was considered separately and its root was set to the node that connected it with the MB. The MB was then removed from further analysis. Considering assumption four we ordered the branches of every subtree according to their length using the `BLO_c1_tree` function. This new *TREES Toolbox* function returns the branch length order (*blo*) values for each branch by first taking the longest path from the root of the subtree and defining it as $blo = 1$. It then defines all the longest paths that branch off from this initial path and labels them as $blo = 2$. This procedure is recursively executed for higher order branches that sprout from previously ordered branches until all branches are labelled (see *Figure 3*). This method was chosen to better accommodate the traditional identification of primary, secondary and tertiary branches in this system. It distinguishes itself fundamentally from the branch order that increases in

steps of one at every branch point away from the root as well as from the Strahler order where order one starts at the dendrite's terminals.

Finally, the angles and curvature values of all nodes of all the subtrees were computed using the new *TREES Toolbox* function `perpendicularity_c1_tree`. Every angle was computed using the inverse tangent (`atan`, *Matlab* function) between two contiguous connected nodes. In addition, the curvature of each node was calculated as described earlier once the angles were computed.

## Morphometrics

A collection of 49 branching statistics was calculated for each dendrite reconstruction separately using a number of different *TREES Toolbox* functions aggregated in our new `features_c1_tree` function. In the following, we enumerate and briefly describe the branching statistics, ordered as found in the `features_c1_tree` code:

1. **Number of branch points** as the sum of all branch points `sum (B_tree (tree))`.
2. **Maximal branch order** as the maximal branch order value of each node in the dendrite. The branch order starts at 0 at the root of the tree and increases after every branch point `max (BO_tree (tree))`.
3. **Mean branch order** as `mean (BO_tree (tree))`. Since the trees were resampled to have one and 0.1 μm distances between nodes each branch order value was thereby approximately weighted by the length of dendrite with that branch order.
4. **Standard deviation of the branch order** as `std (BO_tree (tree))`.
5. **Minimal branch order** of terminals using `BO_tree` and `T_tree`.
6. **Mean branch order of terminals** using `BO_tree` and `T_tree`.
7. **Standard deviation of the branch order of terminals** using `BO_tree` and `T_tree`.
8. **Mean Van Pelt asymmetry index**, average value over all subtrees of a given dendrite (*Uylings and van Pelt, 2002*, `asym_tree`, option: -v).
9. **Standard deviation of the Van Pelt asymmetry index** using (`asym_tree`, option: -v).
10. **Total dendrite length** as the sum of all internode distances, `sum (len_tree (tree))`.
11. **Mean diameter** average node diameters of a given tree, `mean (tree.D)`, after sampling the internode distances.
12. **Standard deviation of the diameter** as `std (tree.D)`.
13. **Mean tapering ratio at branch points** as the standard deviation of the ratio of the diameters between parent and daughter nodes at branching points of a given tree. This is was obtained by combining the `B_tree` function to identify the branching points and `ratio_tree` function to compute the ratios.
14. **Standard deviation of the tapering ratio at branch points** as the standard deviation of the ratio of the diameters between parent and daughter nodes at branching points of a given tree. This was obtained by combining the `B_tree` function to identify the branching points and `ratio_tree` function to compute the ratios.
15. **Total membrane surface** as the sum of the surface in $\mu m^2$ of all segments in a given tree, `sum (surf_tree (tree))`.
16. **Total volume** as the sum of the volume in $\mu m^3$ of all segments in a given tree, `sum (vol_tree (tree))`.
17. **Mean isoneuronal distance of terminals** computed for each tip of a given terminal as the average distance in $\mu m^2$ from that tip to all other nodes in the tree that did not belong to its path to the root. (`isoneuronal_tree`).
18. **Minimal isoneuronal distance of terminals** as the simple average of all of the shortest distances in $\mu m^2$ between terminals and the remaining nodes that did not belong to the same path to the root as the respective terminal of a given tree (`isoneuronal_tree`).
19. **Maximal Euclidean distance to the root** as the maximal euclidean distance in μm between all nodes of the tree and the root (`eucl_tree`).
20. **Mean Euclidean distance to the root** as the mean Euclidean distance in μm between all nodes of the tree and the root (`eucl_tree`).
21. **Standard deviation of Euclidean distance to the root** as the standard deviation of the euclidean distance in μm between a node of the tree and the root (`eucl_tree`).
22. **Mean Euclidean compactness** as the average of the ratios between the Euclidean distance to the root of all nodes and the branch order of the respective node plus one. This was obtained by combining the `eucl_tree` function to calculate the distances between all nodes to the root and `BO_tree` function to find the branch order of the given tree.

23. **Standard deviation of the Euclidean compactness** using `eucl_tree` and `BO_tree`, see above.
24. **Maximal path distance to the root** as the longest metric path length of any node to the root in μm, `max (Pvec_tree (tree))`.
25. **Mean path distance to the root** as the average of the metric path length of all nodes to the root in μm, `mean (Pvec_tree (tree))`.
26. **Standard deviation of path distance to the root** as `std (Pvec_tree (tree))`, in μm.
27. **Mean path compactness** as the average for all nodes of the ratios between the path to the root and the branch order plus one. This was obtained by combining the `Pvec_tree` function to calculate the distances between all nodes to the root and `BO_tree` function to find the branch order of the given tree.
28. **Standard deviation of the path compactness** again using `path_tree` and `BO_tree`.
29. **Mean Tortuosity** as the average of the ratios between the path length and the Euclidean length for each branch individually (`turt_c1_tree`). The branches were defined according to the branch length ordering scheme.
30. **Standard deviation of the tortuosity** using (`turt_c1_tree`).
31. **Mean branching angle** as the average of the angles of all branching points of a tree. An angle was defined as the branching angle within the branching plane between the two daughter nodes of a given branching point (`angleB_tree`).
32. **Standard deviation of the branching angle** using (`angleB_tree`).
33. **Surface of spanning field** as the 2D spanning field in μm$^2$ of the tight contour of a given tree (`span_tree`).
34. **Cable density** was calculated as the ratio between the total length and the surface area of a given tree. This was obtained by combining the `len_tree` function to calculate the total length and the `span_tree` function to calculate the surface area of the tree.
35. **Space-filling** quantifying the efficiency of coverage (*Baltruschat et al., 2020*) of available surface area for a certain dendritic cable length of a given tree using `theta_tree`.
36. **Dendritic field width** as the width of the bounding box around a given tree (`PB_c1_tree`).
37. **Dendritic field height** as the height of the bounding box around a given tree (`PB_c1_tree`).
38. **Dendritic field ratio** as the ratio between the width and height of the bounding box around a given tree (`PB_c1_tree`).
39. **MB ratio** as the ratio between the MB of the c1vpda sensory neurons and the length of the bounding box around a given tree (`PB_c1_tree`).
40. **Total number of terminals** as `sum (T_tree (tree))`.
41. **Terminals lateral density** as the ratio between the number of terminals and the height of the bounding box around a given tree divided by two. This was obtained by combining the `T_tree` function to calculate the number of terminals and the `PB_c1_tree` function to calculate the height of the bounding box around a given tree.
42. **Perpendicularity of lateral branches** as the average angle of all segments of the lateral branches of a given tree (using `perpendicularity_tree`).
43. **Minimal branch length** using `perpendicularity_tree`. The branches were defined according to the branch length ordering scheme.
44. **Mean branch length** using `perpendicularity_tree`.
45. **Standard deviation of branch length** using `perpendicularity_tree`.
46. **Maximal branch length** using `perpendicularity_tree`.
47. **Minimal length over radius ratio** for all segments in a given dendrite as `min(tree.D./len_tree(tree))`.
48. **Maximum length over radius ratio** as `max(tree.D./len_tree(tree))`.
49. **Scaled length** as the total length of a dendrite after scaling it in 2D to ensure that the it covered a target surface area of 100 μm$^2$ using `scaleS_tree`.

## Time-lapse analysis at single branch resolution during the retraction phase

The terminal and branch points of the retraction dataset ($n = 9$) of c1vpda sensory neurons that underwent the retraction phase were registered using `ui_tlbp_tree` (*TREES Toolbox*), a dedicated user interface as described previously (*Baltruschat et al., 2020*), in order to track branch dynamics between 17.5 hrs − 21.5 hrs AEL. Custom written *Matlab* scripts tracked the terminal branch dynamics across time in 1 hr time intervals. The analysis partitioned the terminal branches into five distinct

groups based on their dynamics between each time interval: newly formed branches, shortening branches, extending branches, retracted branches and stable branches that do not change in length, or the changes were below the resolution of the microscope. A similar branch groups classification was used previously (*Stürner et al., 2019*).

### Retraction simulations *in silico*

As a first attempt in understanding the statistical properties of the retracted lateral branches of c1vpda sensory neurons during embryonic development, we defined multiple schemes of terminals retraction based on evidence from the experimental data we collected, covering the plausible regimes of retraction regulation. The simulations followed the steps described next.

For any specified c1vpda time series ($n = 9$) during retraction, we selected the reconstructions when the number of branch points was maximal, that is, before retraction, and when the number of branch points was minimal after retraction. Afterwards, we computed the difference in number of branch points between the aforementioned trees using the `B_tree` function (*TREES Toolbox*).

Then, using the `B_tree`, `T_tree` and `dissect_tree` functions (*TREES Toolbox*) we generated a set of all 'terminal branches' belonging to a given tree before retraction, defined as the piece of dendrite cable between a given termination point and the immediately preceding branch point on its path to the soma. Afterwards, we removed the same number of branches from the tree as the number of branch points difference, by applying four different retraction schemes:

- Small branches first: in this branching scheme the terminal branches were sorted in ascending order by length using the `len_tree` function (*TREES Toolbox*) and the smaller branches removed first.
- Lower angle branches first: terminal branches were sorted in ascending order by the average orientation angle of all segments of the branches using the `perpendicularity_c1_tree` function (new *TREES Toolbox* function) and the branches with lower angles were removed.
- Higher branch length order first: terminal branches were sorted in descending order accordingly to their branch length order using `perpendicularity_c1_tree` function (new *TREES Toolbox* function) and the branches with highest branch length order were removed.
- Random retraction: this retraction scheme contrasts with a rigid and deterministic sequence of programmed retraction, and replaces it by a stochastic retraction. Terminal branches were selected randomly with a uniform distribution and eliminated accordingly. An average over 100 simulations was used.

These results were then analysed and compared as explained in the Results section.

### Computational dendrite growth model with stochastic retraction

The iterative retraction growth model (`growth_c1_tree`) is an extension of the `growth_tree` function from the *TREES Toolbox*, as described in *Baltruschat et al., 2020*. The retraction model was fit to replicate the morphometrics of real dendritic reconstructions during embryonic differentiation. The model reproduces the growth dynamics of real neurons by iteratively adding new branches on a tree at a given time point to produce the tree in the next time point. An additional retraction step was applied on the synthetic trees generated by this growth function to replicate the retraction phase dynamics of the c1vpda sensory neurons (see *Figure 6—figure supplement 1* for simulations without retraction).

To model a given c1vpda time series that experienced retraction, the algorithm started by selecting the reconstruction when the number of branch points was maximal, that is, before retraction. Then it computed the mean branch rate ($B_r$) of all neurons per time interval (15 mins), between the time point when the imaging experiment started (16 hrs AEL), and the time point before retraction (19.5 hrs AEL). To incorporate the initial main branch (MB) polarisation described in real neurons, the growth was simulated starting with an existing real initial MB. The MB of a given tree was found by applying the function `PB_c1_tree` (new *TREES Toolbox* function) on a selected tree. After stripping the MB from the real morphology, the algorithm extracted the contour of the dendritic spanning field of the initial tree using the function `boundary` (*Matlab* function), with parameter $\alpha = 1$ and positioned the MBs inside the corresponding dendritic field of the tree before retraction. This spanning area defined the geometry where the simulations are performed. The numerical simulations of the model dynamics were performed within the 2*D* physical boundary, enacting the combined effect

of transmembrane and membrane molecules (*Meltzer et al., 2016*; *Kim et al., 2012*; *Han et al., 2012*) that facilitate cell-extracellular matrix adhesion, confining sensory neurons to a 2*D* space.

The noisy growth phase of the model was then initialised and at each iteration the surface area was probed with $N = 100,000$ random target points. For each target point the shortest Euclidean distance to the tree was detected and the resulting distances were capped at a maximal growth range radius of $r = 2.5 \mu m$, before retraction (19.5 hrs AEL) and $r = 1.81 \mu m$ after retraction. These radii were defined as the average growth rate of new branches until and after retraction respectively (from *Figure 5B*). Then, a target point was chosen at random with a preference for points with a larger Euclidean distance (noise parameter $k = 0.5$) to enable space filling. The selected target point was then connected to the closest point on the tree minimising cable length and path length cost with a $bf = 0.2$ as found for the MST model used to test the wire optimisation of the c1vpda dendrites (see model in *Baltruschat et al., 2020*). At each iteration the synthetic trees grew at rate $B_r$, between $16-19.5$ hrs AEL for the case of the retraction models, and between $16-22.5$ hrs AEL for the case of the model without retraction. The simulations stopped when time point 22.5 hrs AEL was reached.

In parallel with the noisy growth step, the model entered a phase of dynamic retraction at time points 16.5, 17.5, 18.5, 19.5, 20.5, 21.5 hrs AEL, taking into account the 1 hr resolution of the time-lapse data. Evidence from the single branch tracking data was used to constrain the model retraction steps. The retraction rate and distribution of branches per class data was then divided and averaged into bins with the corresponding bin edges: $\leq 17.5 \leq 18.5 \leq 19.5 \leq 20.5 \leq 21.5$ (*Figure 5B*). At each of the aforementioned time points, terminals are selected at random for their tips to be shortened. The percentage of branches selected for shortening was defined as the combined percentage of retracting and shortening branches at the corresponding time bin in the real data. Each tip of the selected terminals is shortened in the same amount as the average cable length of retracted branches found in the real neurons, in that time bin. If the amount of cable to be shortened surpassed the terminal length the branch was removed from the tree. Moreover, a proportion of new branches were added to the existing tree equalling the percentage of newly formed branches at the same time bin in the real data. The simulated results were then analysed and compared with the morphometrics from the real neurons as explained in the Results section.

## Statistical analysis

Statistical tests and all data analysis were performed using *Matlab* (http://www.mathworks.com) and they were implemented in custom-made code. Statistical parameters including the exact value of the sample size and precision measures (mean ± SEM or mean ± SD) are reported in the figures and the text. All statistical evaluations were done empirically by means of bootstrap hypothesis testing to avoid any data distribution assumptions. All *p* values were reported as: * $p<0.05$, ** $p<0.01$, *** $p<0.001$.

## Acknowledgements

We are grateful to A Berthelius for comments on the manuscript. We would like to thank M Weigand for help with using the 3D printer, and to A Kohli and R Khamatnurova for discussions on the tree alignment algorithm. This work was supported by a BMBF grant (No. 01GQ1406 – Bernstein Award 2013 to HC), by University Medical Center Giessen and Marburg (UKGM) core Funding (to PJ), by DZNE core Funding (to GT) and by a DFG grant (SPP 1464 to GT). The authors declare to have no competing financial interests.

## Additional information

### Funding

| Funder | Grant reference number | Author |
| --- | --- | --- |
| Bundesministerium für Bildung und Forschung | 01GQ1406 | Hermann Cuntz |
| Deutsche Forschungsgemeinschaft | SPP 1464 | Gaia Tavosanis |

| University Medical Center Giessen and Marburg (UKGM) | Core Funding | Peter Jedlicka |
| Deutsches Zentrum für Neuro-degenerative Erkrankungen | Core Funding | Gaia Tavosanis |
| Bundesministerium für Bildung und Forschung | 031L0229 | Peter Jedlicka |

The funders had no role in study design, data collection and interpretation, or the decision to submit the work for publication.

### Author contributions

André Ferreira Castro, Conceptualization, Data curation, Investigation, Visualization, Methodology, Writing - original draft, Writing - review and editing, performed experiments, data analysis, modelling; Lothar Baltruschat, Conceptualization, Investigation, Writing - review and editing, performed experiments; Tomke Stürner, Conceptualization, Writing - review and editing; Amirhoushang Bahrami, Conceptualization, Formal analysis, Writing - review and editing; Peter Jedlicka, Conceptualization, Supervision, Funding acquisition, Writing - review and editing; Gaia Tavosanis, Conceptualization, Resources, Supervision, Funding acquisition, Writing - original draft, Project administration, Writing - review and editing; Hermann Cuntz, Conceptualization, Supervision, Funding acquisition, Investigation, Visualization, Writing - original draft, Project administration, Writing - review and editing, performed modelling

### Author ORCIDs

André Ferreira Castro https://orcid.org/0000-0002-6841-1952
Tomke Stürner https://orcid.org/0000-0003-4054-0784
Amirhoushang Bahrami http://orcid.org/0000-0001-5841-2516
Peter Jedlicka https://orcid.org/0000-0001-6571-5742
Gaia Tavosanis https://orcid.org/0000-0002-8679-5515
Hermann Cuntz https://orcid.org/0000-0001-5445-0507

### Decision letter and Author response

Decision letter https://doi.org/10.7554/eLife.60920.sa1
Author response https://doi.org/10.7554/eLife.60920.sa2

## Additional files

### Supplementary files

• Transparent reporting form

### Data availability

All data and all code is available on Zenodo https://doi.org/10.5281/zenodo.4290200.

The following dataset was generated:

| Author(s) | Year | Dataset title | Dataset URL | Database and Identifier |
|---|---|---|---|---|
| Castro FA, Baltruschat L, Stürner T, Bahrami A, Jedlicka P, Tavosanis G, Cuntz H | 2020 | Dataset for Achieving functional neuronal dendrite structure through sequential stochastic growth and retraction | https://doi.org/10.5281/zenodo.4290200 | Zenodo, 10.5281/zenodo.4290200 |

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

# Appendix 1

## In brief

An optimal wire and function trade-off emerges from noisy growth and stochastic retraction during *Drosophila* class I ventral posterior dendritic arborisation (c1vpda) dendrite development.

Highlights

- C1vpda dendrite outgrowth follows wire constraints.
- Stochastic retraction of functionally suboptimal branches in a subsequent growth phase.
- C1vpda growth rules favour branches running parallel to larval body wall contraction.
- Comprehensive growth model reproduces c1vpda development *in silico*.

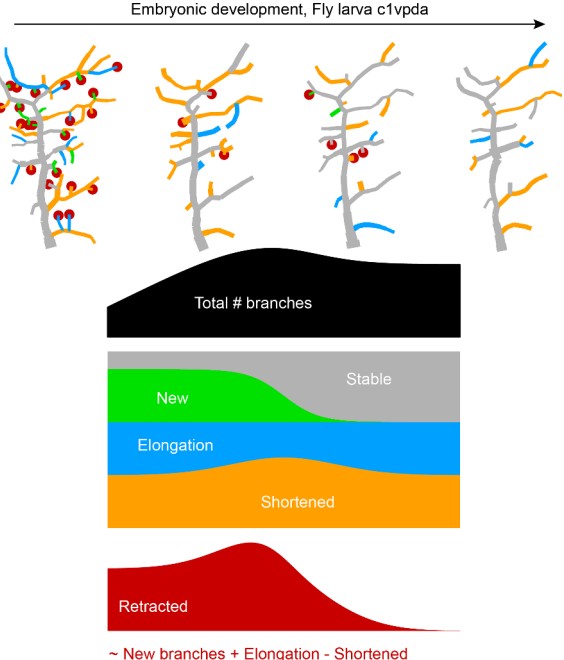

**Appendix 1—figure 1.** Sketch that illustrates visually the dynamics of branching during embryonic development of class I dendrites.

