## [Decision Letter]

**Acceptance summary:**

Neuronal morphology is diverse and highly variable across neuron types, within and across species. A principled hypothesis for understanding this diversity is that it reflects differing functional requirements and developmental constraints. This study focuses on the development of the dendritic structure of mechanosensory neurons in *Drosophila* larvae, providing a detailed quantification of how dendritic branches elaborate and retract. A family of plausible mathematical models for growth reveals that a parsimonious stochastic retraction mechanism can explain the developmental profile of these complex dendritic structures. Both the data and modelling insights will be of interest to a broad section of developmental biologists, anatomists and neuroscientists.

**Decision letter after peer review:**

Thank you for submitting your article "Achieving functional neuronal dendrite structure through sequential stochastic growth and retraction" for consideration by *eLife*. Your article has been reviewed by three peer reviewers, one of whom is a member of our Board of Reviewing Editors, and the evaluation has been overseen by K VijayRaghavan as the Senior Editor. The following individual involved in review of your submission has agreed to reveal their identity: Athanasia Papoutsi (Reviewer #3).

The reviewers have discussed the reviews with one another and the Reviewing Editor has drafted this decision to help you prepare a revised submission.

Summary

This work gathers and analyses quantitative morphological data of the development of *Drosophila* sensory neurons during larval development. The manuscript documents growth and retraction of neurites, giving an overall picture of how dendrites on a specific class of proprioceptors develop and how the developmental process may optimise the shape of the dendritic tree to sense bending. The authors propose a parsimonious stochastic optimization model for how neurites might grow, subject to constraints such as wiring cost. This model is found to give a good account of the data.

The reviewers raised several points that the authors should address in a revision of the paper. These are summarised here; the full reviewer comments are included below for reference and need not be addressed point-by-point. The reviewers agreed that no further experiments are necessary.

1) Check/provide evidence that the sampling rate of the time lapse imaging doesn't alias the measurements of dendritic growth and/or present a problem for relating data to the model.

2) Small variations in the model were identified – how are these justified (Figure 6B vs 2D)?

3) The authors should consider balancing their citations toward invertebrate quantitative neuroanatomy and existing quantitative/computational development studies.

4) Additional data for Figure 5A (as a figure supplement) would aid intuition.

Reviewer #1:

This study gathers and analyses quantitative morphological data of the development of *Drosophila* sensory neurons during larval development. They document growth and retraction of neurites in a very accessible and comprehensive way, giving a nice overall picture of how dendrites on a specific class of proprioceptors develop. The authors propose a parsimonious stochastic optimization model for how neurites might grow, subject to constraints such as wiring cost. This model is found to give a good account of the data.

The manuscript is well presented and methodologically sound. The conclusions are justified. The study should be of broad interest to developmental anatomists, theoretical biologists, neurophysiologists and the connectomics community.

It could be published as-is but would possibly benefit from more balanced and carefully curated scholarship: a lot of the citations in the Introduction concern mouse/rat neurophysiology, neglecting a lot of invertebrate studies. It is nice to connect the findings to dendritic function, but I feel the emphasis on dendritic physiology (e.g. computation in the visual system) is not balanced fairly against the (huge) amount of recent quantitative neuroanatomy work done in this very organism. Similarly, there is an omission of a lot of quantitative developmental work (both modelling and experiment, going back decades). Finally, there are a number of tangential studies that seem to be cited just to bulk up the references. I think the paper will be better received and will benefit the field more if the references could be rebalanced a little, should the authors wish to do so.

Reviewer #2:

The complexity of dendrite development has so far made it difficult to quantitatively describe their growth accurately. This work features in vivo imaging of a *Drosophila* sensory neuron called vpda, together with a convincing computational model that accurately describes most features of its development. In this aspect, the work is very interesting and poses a significant advance in regard to the quantitative description and understanding of dendritic tree growth and refinement during development. It further considers the functional aspects of vpda, based on curvature analysis and optimization of dendrite orientation for its proprioceptive function. Overall, I believe this is very interesting and thorough work shedding new insight into the little understood rules of dendritic tree development.

1) The embryonic imaging of vpda neurons was performed at 5min intervals, but quantifications were done only at 0.5-1h intervals. This might severely underestimate the growth dynamics during this stage, as dendrites can grow several microns within few minutes. Yet most of the authors' assumptions are based on this analysis. I think in particular growth and retraction rates might be potentially misrepresented, as both can happen to the same branch during the analyzed 30-60 min time intervals. The authors should at least consider this point and make a statement in this regard, or ideally validate their assumptions by analyzing their data at higher temporal resolution.

2) A very complementary preprint (Palavalli et al., 2020) describing and modelling vpda growth arrives at a similar conclusion as the work presented here. However, the computational models differ. Unlike here, the other model considers contact based self-repulsion (mediated by Dscam) as a key feature, resulting in similar simulated dendrite morphologies. I feel it is intriguing that both models basically arrive at the same result, yet the underlying assumptions differ. In my mind, this suggests that different growth mechanisms can result in similar outcomes. Alternatively, self-avoidance is not the only mechanism contributing to 2nd order branch parallelization, but acts in concert with stochastic retraction as described here. As dendrite self-avoidance is nonetheless a major feature of dendrite development from a biological point of view, likely across species and most neurons, the authors should discuss this point more carefully. Can they speculate about a biological correlate for the assumptions they make in their model, which results in stochastic retraction? As most of the parameters of the authors' model are directly derived from experimental data, this might inadvertently include the requirements for self-avoidance.

3) The authors show a very interesting computational analysis of dendrite curvature during larval locomotion, which coincides with neuronal calcium responses. While this result is not entirely novel as stated by the authors (see Vaadia et al., 2019), it still implies that dendrite development is adjusted to optimize its purpose. It is intriguing that the retraction of dorsoventral vpda dendrites coincides with peristaltic activity of the developing animal during the late embryonic stage. For the motor system, a critical period has been postulated around 17-19 h AEL, where rhythmic peristalsis and coordinated activity of motor neurons emerges. Have the authors considered a role for activity in reshaping of the dendritic tree during this phase, particularly during the retraction phase? It would be fairly straight forward to test this by Kir2.1-mediated silencing of vpda, providing a biological correlate shaping the final dendritic morphology.

Reviewer #3:

In the submitted manuscript, Castro and colleagues investigate the development of the dendritic tree of the c1vpda neurons of the *Drosophila*, by using a combination of in vivo imaging techniques and modelling of synthetic morphologies. The main conclusion of this study is that, during embryonic development, dendritic growth follows the optimal wire principle and that the subsequent stochastic retraction of dendritic branches allows for the maturation of a functional dendritic tree. Overall, the paper describes in detail this developmental process and is of potential wider interest as it showcases how stochastic processes can result in specialized dendritic tree patterns. I also appreciate the authors making their data and code available for the review process. I support this work for publication in *eLife*, yet I have some comments that would increase the clarity and visibility of the author's results.

1) My main concern has to do with the computational model shown in Figure 6, in combination with the model in Figure 2D/Figure 6B, dashed line. Specifically, I found it confusing why the authors chose to implement two different approaches to model dendritic growth. The results presented Figure 6B seem to better model the experimental data of Figure 2D, for a low number of branch points. The main differences between the two models, as far as I can tell, are in the surface area used to position the random targets and the inclusion of distinct time points. In other words, how are the two approaches different, and do they convey a principle for the developmental growth of the dendrites?

2) The results shown in Figure 5A are interesting and support the author's claim for the random retraction of dendrites. Can the authors provide information regarding the properties of the terminal branches (distributions of length, angle and BLO) that will make these results more intuitive?

---

## [Author Response]

The reviewers raised several points that the authors should address in a revision of the paper. These are summarised here; the full reviewer comments are included below for reference and need not be addressed point-by-point. The reviewers agreed that no further experiments are necessary.1) Check/provide evidence that the sampling rate of the time lapse imaging doesn't alias the measurements of dendritic growth and/or present a problem for relating data to the model.

Using the tools and resources available to us at this point, a higher temporal resolution analysis would not be feasible. This would require hundreds of additional reconstructions and their registration for the time-lapse analysis. The analysis is therefore bound to the resolution that we used and we agree with reviewer #2 (major comment #1) that higher frequency dynamics will elude our analysis. However, we would like to note that we used the same temporal resolution in the time-lapse experiments and in the random retraction growth model. The focus of this data-driven model was to replicate the growth behaviour of c1vpda neurons at this temporal scale. The agreement between simulated data and experiments validates the interpretation of the data. Higher resolution analyses will surely be useful in the future to better understand the fine details of the dynamics of dendrite growth in these cells. Nevertheless, the aim of our study was to connect the function of c1vpda neurons to dendrite differentiation across embryonic and larval stages. After the submission of the present manuscript, a complementary pre-print focusing on the molecular basis of embryonic c1vpda neurons development was submitted to bioRxiv (Palavalli et al., 2020). Since this study focused in a much smaller developmental timewindow of analysis, they used higher temporal resolution imaging to provide a link between Dscam1 molecule and branch dynamics. This level of specificity comes with the cost of not capturing the overall functional assembly of c1vpda neurons (see as well major comment #2 from reviewer #2). Finally, in line with the suggestion of the reviewer, in the revised paper (both in Results and Discussion), we put emphasis on the potential differences in growth-retraction branch dynamics that may arise due to different temporal resolutions of imaging:

”Note that the results are the net value of branch tip position between observation points at time intervals of 1*hr*. Even though the interpretation of this analysis is valid for the selected time interval, higher temporal resolution will no doubt uncover higher frequency dynamics that are not captured here, a common problem of any time-lapse analysis and indeed of any type of image processing (Helmstaedter, 2013; Peng et al., 2017). However, the net changes within 1*hr* intervals are a stable result and provide for a phenomenological description and quantification of the branch reorganisation throughout the retraction phase.”;

”Due to the lack of high-throughput automatic digital tracing algorithms for neuronal morphologies the chosen temporal resolution for imaging (see Materials and methods) was determined as a trade-off between capturing c1vpda dendrite dynamics throughout embryonic and larval stages and tracing speed (Helmstaedter, 2013).”;

and

”The accordance between data and model further validates the chosen temporal resolution of imaging. In the future, as more sophisticated tracing algorithms develop more detailed datasets will become available for analysis and modelling (Peng et al., 2017).”.

Furthermore, we will make all the raw image-stacks available upon publication. This way, by providing all data at temporal resolution of 5*min* intervals, readers interested in this topic can trace the image-stacks and analyse the data in more detail.

2) Small variations in the model were identified – how are these justified (Figure 6B vs 2D)?

Thank you for highlighting this point. To improve clarity we added new text in the Results section and Figure 6 legend with further details about the differences between the models. Please see below some points that clarify this comment:

1) The model in Figure 2D/Figure 6B (MST model – represented as a dashed line in those figures) is a reconstruction model. Reconstruction algorithms are the simplest of dendrite patterning computational models. They do not capture the evolution of the developmental processes and simply reconstruct dendrites at a particular point in time. These algorithms are designed to generate morphologies that replicate the shape of real cells at a static developmental time point. First, from a given dendritic tree different morphometrics are quantified, e.g. branch points, length, surface area. Then, the modeller makes assumptions about how these variables relate between each other. Particularly in the MST model used in our study, it is assumed that wiring minimisation is a key principle of dendrite wiring. Based on these assumptions, the algorithm then proceeds to generate morphologies by sampling from the distributions of real data.

2) The newly developed random retraction growth model (Computational dendrite growth model with stochastic retraction – in Materials and methods) seeks to explain the spatiotemporal differentiation of dendritogenesis. Such developmental algorithms need to specify how distinct morphometrics evolve through time, in order to make predictions about the growth process, as showed in Figure 6C and 6D. To properly constrain this model we needed dynamical morphometrics from the time-lapse data to specify how the distributions of the parameters change over time: such as elongation, branching and retraction rate of dendritic branches.

3) In Figure 2D, we first used the simpler static MST model as a first approach to test if the real data followed wiring minimisation principles. For each c1vpda morphology a MST counterpart with similar morphometrics was generated for comparison. After we developed the new random retraction growth model we used it to generated synthetic morphologies. Then, to validate random retraction growth model we wished to verify if the new synthetic dataset followed the same trends as the real data. As a first step, in Figure 6B, we compared the synthetic trees generated by the random retraction growth model with the MST model used in Figure 2D. The very good agreement between synthetic data generated by the random retraction growth model and the MST dashed line provided a first quality control for the data produced by our newly developed model. From Figure 2D/6B we gained the insight that real data and synthetic trees generated by the random retraction growth model behave similarly in respect with wiring optimisation.

3) The authors should consider balancing their citations toward invertebrate quantitative neuroanatomy and existing quantitative/computational development studies.

Thank you for the comments. We followed the suggestion of the reviewer and added new references to the Introduction. Particularly, we added references to recent studies in quantitative neuroanatomy during development. Additionally, we added references on the impact of dendritic morphology on computation in insects.

We reorganised the references in the Introduction to better highlight the importance of neuroanatomical studies in invertebrates. Note that the field of quantitative neuroanatomy is experiencing a revolution at the moment and citing all the published work and pre-prints available would be more suitable for a review paper. Finally, we added references to the Discussion section of the manuscript that focus on neuronal cell fate specification.

4) Additional data for Figure 5A (as a figure supplement) would aid intuition.

Thank you for your comment. The different simulations in the paper have different goals and address different questions. In Figure 5A (now Figure 5—figure supplement 1), we provide a first attempt at disentangling the morphological implications of the retraction phase by simulating the most extreme cases of branch retraction. It is not the point of the analysis performed in this figure to fully reproduce the different morphometric distributions in great detail. It was done only to provide insights for the next round of in-depth investigations presented in Figure 6. Therefore we restricted our statistical tests to mean comparison between key morphometrics. To avoid confusion for the reader we opted to move Figure 5A to Figure 5—figure supplement 1 that supports Figure 5 in the main text.